# DMol: A Highly Efficient and Chemical Motif-Preserving Molecule Generation Platform

**Peizhi Niu[1], Yu-Hsiang Wang[1], Vishal Rana[1],**
**Chetan Rupakheti[2], Abhishek Pandey[2], Olgica Milenkovic[1]**
[1]University of Illinois Urbana-Champaign
[2]AbbVie
{peizhin2, yw121, vishalr, milenkov}@illinois.edu
{chetan.rupakheti, abhishek.pandey}@abbvie.com

## Abstract

We introduce a new graph diffusion model for small drug molecule generation which simultaneously offers a 10-fold reduction in the number of diffusion steps when compared to existing methods, preservation of small molecule graph motifs via motif compression, and an average $3\%$ improvement in SMILES validity over the DiGress model across all real-world molecule benchmarking datasets. Furthermore, our approach outperforms the state-of-the-art DeFoG method with respect to motif-conservation by roughly $4\%$, as evidenced by high ChEMBL-likeness, QED and newly introduced shingles distance scores. The key ideas behind the approach are to use a combination of deterministic and random subgraph perturbations, so that the node and edge noise schedules are codependent; to modify the loss function of the training process in order to exploit the deterministic component of the schedule; and, to "compress" a collection of highly relevant carbon ring and other motif structures into supernodes in a way that allows for simple subsequent integration into the molecular scaffold[1].

## 1 Introduction

Graphs are fundamental data structures used to model a wide range of complex interactions, including molecules and drugs, biological pathways, and social and co-purchase networks Wu et al. [2021], Zhang et al. [2024], Yang et al. [2024], Waikhom and Patgiri [2021]. Hence, the ability to generate graphs that accurately capture domain-specific distributions is of great significance Tsai et al. [2023], You et al. [2018], Gamage et al. [2020]. While generative models for images and texts are being used with great success Dhariwal and Nichol [2021], Ho et al. [2022], Waikhom and Patgiri [2021], graph generation remains a challenging frontier due to the discrete nature of the models, the need for permutation invariance and other distinctive properties Velikonivtsev et al. [2024], Wang et al. [2023]. This is particularly the case when trying to generate graph samples that innovate molecular compounds Jin et al. [2018], Mitton et al. [2021], Vignac and Frossard [2021], Gómez-Bombarelli et al. [2018], Kwon et al. [2020], Wu et al. [2024], You et al. [2018], Liao et al. [2019], Segler et al. [2018], Popova et al. [2019], Madhawa et al. [2019], Zang and Wang [2020], in which case the generated samples can fail to obey necessary biochemical and physical constraints and preserve important motif compounds.

One way to address these challenges is to use diffusion models Sohl-Dickstein et al. [2015], Ho et al. [2020], Fan et al. [2023], Chen et al. [2024], Trippe et al. [2023], Haefeli et al. [2023], Niu et al. [2020b], Huang et al. [2022], Liu et al. [2024c] which rely on a gradual noise injection process that perturbs the input samples and then reverses the process using a learned denoising network.

---

[1]The link to the code can be found at: https://github.com/liekon/Discrete-Graph-Generation

In the context of small molecule drug generation, existing methods focus on *discrete diffusion* in order to avoid the loss of key structural properties that arise when embedding graph structures into continuous spaces Niu et al. [2020a], Jo et al. [2022b]. Discrete diffusion models incorporate "transition mechanisms" that explicitly consider the categorical attributes of nodes and edges, ensuring fine-grained structural changes during the noise injection process Liu et al. [2024b], Xu et al. [2024a]. They also allow for continuous and discrete denoising steps that diversify the outputs Huang et al. [2022], Vignac et al. [2023], Jo et al. [2022a]. However, they also typically require a very large number of diffusion steps, resulting in large running times and extremely slow graph generation Kong et al. [2023]. More importantly, they lead to structures that have low chemical validity even when measured through exceptionally coarse SMILES nomenclature constraints Hoogeboom et al. [2022], Liu et al. [2021]. Most learning methods consequently fail to produce molecules that can be chemically synthesized or that can properly dock on target proteins.

One well-known graph diffusion method, DiGress Vignac et al. [2023], relies on a cosine noise injection mechanism that controls the amount of perturbations during the different diffusion steps. In the process, every node and edge in the graph is allowed to be modified at each step. DiGress evaluates the generated drug sample quality through SMILES validity, novelty, and uniqueness all of which provide a highly limited measure of the utility of molecules. Also, the training and sampling time of the method is excessive. Furthermore, in its current form, and unlike some nondiffusion based methods Jin et al. [2018], Simonovsky and Komodakis [2018], Mitton et al. [2021], Vignac and Frossard [2021], Gómez-Bombarelli et al. [2018], Kwon et al. [2020], Wu et al. [2024], You et al. [2018], Liao et al. [2019], Segler et al. [2018], Popova et al. [2019], DiGress cannot preserve the structures and frequencies of important drug network motifs (including carbon rings, which are the most important building blocks of organic molecules).

We describe a new small molecule drug diffusion model, Diffusion Models for Molecular Motifs, **DMol**, which significantly outperforms DiGress across all benchmarking datasets in terms of running time, SMILES validity, and novelty scores and at the same time, allows for network motif preservation via compressed representations of subsets of motifs. At the same time, the DMol improves the more informative chemical validity scores when compared to the very recent state-of-the-art DeFoG Qin et al. [2025] method by roughly $4\%$, since the latter cannot guarantee preservation of motifs. Its main features are as follows:

**1.** DMol directly relates the **diffusion time-steps** with the **number** of nodes and edges that are allowed to undergo state changes in that step. This approach effectively increases the learning rate of the model as it learns increasingly larger submodels in a gradual manner. It also significantly reduces the time required for molecule generation, leading to an order of magnitude reduction in the number of steps required by diffusion models.

**2.** The **number** of nodes and edges that DMol perturbs at each step is **deterministic**, but the nodes themselves are selected at random. Furthermore, the perturbed edges are confined to the subgraph induced by the selected vertices, which couples the node and edge perturbations and leads to better preservation of subgraph structures.

**3.** DMol uses new penalty terms in the objective function that **penalize mismatches** in the deterministic counts of nodes and edges perturbed during the forward and backward steps. Such penalties lead to poor results when the number of perturbed nodes is random, as in DiGress and other methods, and motivates the use of a mixed deterministic/random noise schedule. This allows for further "forced motif" preservation and a more flexible control of the noise schedule/distribution, as evidenced by a straightforward theoretical analysis.

**4.** One of the most important features of DMol is **motif compression** which allows one to convert chemically important node motifs into supernodes that are diffused similarly to regular atom nodes. The chosen motifs (which are mostly carbon rings) have the property that they can always be reattached to the molecular scaffold, and the number of such molecules is bounded to retain computational savings guaranteed by the first three innovations. Due to their careful chemical selection, only one type of bond is possible during reconstruction, and "decoding" is extremely simple. Motif compression differs both from JT-VAEs Jin et al. [2019] and ring freezing proposed in DiGress (which failed to perform in a satisfactory manner), since in our case one is still allowed to **directly** replace one valid motif structure by a single atom or another valid motif structure (in addition, the choices of substructures used differ widely).

As a result, DMol improves the SMILES validity of DiGress by roughly 3% over all real-world molecular benchmarking datasets while keeping the number of diffusion steps at an order that is one magnitude smaller. At the same time, the significantly more meaningful chemical likeness scores (e.g., RDKit's Landrum et al. [2006] ChEMBL likeness, QED and shingle distances, which indicate how likely the generated molecule is to have desired chemical motifs and similar biological activity properties) of DMol are consistently better than those of DiGress. These scores for DMol are roughly 4% higher, on average, than those of the recent state-of-the-art flow matching method DeFoG Qin et al. [2025], since the latter does not tend to preserve chemical motifs.

As a final remark, it is important to point out that in the biochemistry literature, it has been long recognized that SMILES validity optimization may not be as practically relevant from the chemical synthesis/properties point of few; motif, and especially carbon ring structure preservation, is deemed much more important, but tangible evaluation metrics for true chemical validity are still lacking. See for example the recent work Skinnider [2024] that discusses pros and cons of SMILES validity.

## 2 Related Works

Viewing small molecules as graphs has inspired a myriad of generative models for drug design, including JT-VAE Jin et al. [2018], MoFlow Zang and Wang [2020], CDGS Huang et al. [2023], EDM Hoogeboom et al. [2022], etc Cornet et al. [2024], Shi et al. [2021], Xu et al. [2024a], Chen et al. [2023], Wu et al. [2022], Bao et al. [2023], You et al. [2018], Liao et al. [2019], Mercado et al. [2021], Segler et al. [2018], Kwon et al. [2020], Jensen [2019], Jo et al. [2024].

Furthermore, numerous surveys on the subject are readily available Tang et al. [2024], Du et al. [2022], Yang et al. [2024]. We partition and review prior works on generative models for drug discovery based on the methodology used.

**Traditional Methods.** Traditional approaches to graph-based molecular generation relied on the use of Variational Autoencoders (VAEs) as generative models Jin et al. [2018], Simonovsky and Komodakis [2018], Mitton et al. [2021], Vignac and Frossard [2021], Gómez-Bombarelli et al. [2018], Kwon et al. [2020], Wu et al. [2024]. VAEs comprise an encoder that maps the input graph into a latent space and a decoder that reconstructs the graph from the latent embedding. Many methods also involve autoregressive models, which generate molecular graphs by sequentially predicting the next element in the sequence based on previous outputs You et al. [2018], Liao et al. [2019], Segler et al. [2018], Popova et al. [2019]. On the other hand, flow-based methods leverage the concept of normalizing flows for graph generation Madhawa et al. [2019], Zang and Wang [2020], Luo et al. [2021b], Lippe and Gavves [2021], Shi* et al. [2020]. These techniques apply a series of invertible transformations to model complicated distributions based on simpler ones (such as a Gaussian). Additionally, Generative Adversarial Networks (GANs) approaches have also been used for molecule generation De Cao and Kipf [2018], Łukasz Maziarka et al. [2019], Martinkus et al. [2022a]. GANs include a generator, which learns to create realistic molecular graphs, and a discriminator, which learns to differentiate between real and generated samples.

**Diffusion-Based Methods.** Diffusion models for graph generation can be broadly categorized as follows. Denoising Diffusion Probabilistic Models (DDPM) utilize a Markov chain for the diffusion process Vignac et al. [2023], Xu et al. [2022], Liu et al. [2024b], Xu et al. [2024a], Liu et al. [2024a], Hoogeboom et al. [2022], Jo et al. [2024]. Score-Based Generative Models (SGM) leverage score matching techniques to model the data distribution, and generate graphs by reversing a diffusion process guided by the score of the data distribution Luo et al. [2021a], Chen et al. [2023], Wu et al. [2022]. Furthermore, by replacing discrete time steps with continuous time, one can use stochastic differential equations to model the diffusion and denoising processes Jo et al. [2022a], Lee et al. [2023], Huang et al. [2023], Bao et al. [2023]. Furthermore, studies have shown that different noise schedules may significantly affect molecule generation quality Shi et al. [2025], Nichol and Dhariwal [2021]. Comprehensive surveys on the subjects include Mengchun Zhang et al. [2023], Fan et al. [2023], Chen et al. [2024].

DiGress Vignac et al. [2023] uses a denoising diffusion probabilistic model within a discrete state space and is considered one of the practically effective models for molecular generation tasks. It represents graphs in the discrete space of node and edge attributes by assigning categorical labels to each node and edge. DiGress models the noise addition process as a Markov process, where each node and edge label evolves independently of the others. This assumption mirrors the approach in image-based diffusion models. Denoising is performed using a graph transformer network, trained by

minimizing the cross-entropy loss between the predicted probabilities for nodes and edges and the ground-truth graph. The transformer network reconstructs a denoised graph from a noisy input. To generate new graphs, a noisy graph is first sampled according to the model's limiting distribution, and the trained denoising transformer is then used to produce a graph with the desired properties. The DiGress model is discussed further in Section B.

DisCo Xu et al. [2024b] uses discrete-state continuous-time diffusion, addressing limitations of discrete-time approaches like DiGress. The model formulates graph generation as a continuous-time Markov Chain (CTMC) that preserves the discrete nature of graph-structured data while enabling flexible sampling trade-offs. Unlike discrete-time models with fixed sampling steps, DisCo can adjust sampling steps after training without retraining the model. DisCo enables numerical approaches like $\tau$-leaping for efficient simulation of the reverse process, allowing practitioners to dynamically balance generation quality and computational efficiency during inference. Despite its innovations, DisCo suffers from quadratic computational complexity with respect to the number of nodes, limiting its scalability for generating large graphs. Moreover, DisCo reduces the number of sampling steps at the cost of degraded generation quality.

**Flow Matching-based Methods.** Flow Matching (FM) has recently emerged as an alternative to diffusion models for generative tasks Lipman et al. [2023], Dao et al. [2023]. FM frameworks define continuous probability paths between a simple prior distribution and the target data distribution, offering comparable performance and efficiency compared to diffusion-based approaches. To address discrete state spaces, Discrete Flow Matching (DFM) formulations have been developed Gat et al. [2024], providing a streamlined approach with more flexible sampling procedures. DeFoG Qin et al. [2025] extends the DFM framework specifically for graph generation tasks. It employs a linear interpolation noising process and a CTMC-based denoising process while preserving the inherent symmetries of graphs. The key innovation in DeFoG is its disentanglement of training and sampling stages, which enables independent optimization of each component. Despite its innovation, DeFoG still requires dataset-specific hyperparameter tuning for optimal sampling strategies, making its adoption potentially challenging for new graph domains without extensive experimentation. Importantly, experiments to be described in this work reveal that DeFoG under-performs on key biochemical motif molecular property metrics such as QED.

## 3 DMol Diffusion

### 3.1 Review of Standard Graph Diffusion Models

The focal concept in diffusion models is the noise model $q$, designed to enable $T$ forward diffusion steps. Graph diffusion relies on progressively introducing noisy perturbations to an undirected graph $\mathcal{G}^0$, encoded as $\mathbf{z}^0$, resulting in a sequence of increasingly noisy data points $(\mathbf{z}^1, \ldots, \mathbf{z}^T)$. The noise addition process is Markovian, which translates to $q(\mathbf{z}^1, \ldots, \mathbf{z}^T \mid \mathbf{z}^0) = \prod_{t=1}^{T} q(\mathbf{z}^t \mid \mathbf{z}^{t-1})$.

More precisely, the encoding of a graph involves categorical information about the nodes $V$ and vertices $E$ of the graph $\mathcal{G} = (V, E)$. The assumptions are that each node belongs to one of $a$ classes and is represented as a one-hot vector in $\mathbb{R}^a$. The same is true for edges, which are assumed to belong to one of $b$ classes (with one class indicating the absence of the edge). We follow the notation from Vignac et al. [2023], where $\mathbf{x}_i$ is used to denote the one-hot encoding of the class of node $i$. The one-hot encodings are arranged in a node matrix $\mathbf{X}$ of dimensions $n \times a$. Again, in a similar manner, edges are encoded into a tensor $\mathbf{E}$ of dimension $n \times n \times b$. The Markovian state space for each node and edge is the set of classes of the nodes and edges, respectively. With a slight abuse of notation, $\mathbf{x}_l$ stands for the class (state) of node $l$, while $\mathbf{e}_{ls}$ stands for the class (state) of edge $ls$.

Diffusion is performed separately on nodes and edges, while the noisy perturbations in the forward process are governed by a state transition matrix. For node-level noise we use transition matrices $(\mathbf{Q}_X^1, \ldots, \mathbf{Q}_X^T)$, where each entry $[\mathbf{Q}_X^t]_{ij}$ defines the probability of an arbitrary node transitioning from state $i$ to state $j$ at time step $t$. A similar model is used for edges, where the transition matrices $(\mathbf{Q}_E^1, \ldots, \mathbf{Q}_E^T)$ describe the probabilities $[\mathbf{Q}_E^t]_{ij}$ of an arbitrary edge transitioning from state $i$ to state $j$. Formally, $[\mathbf{Q}_X^t]_{ij} = q(x^t = j \mid x^{t-1} = i)$ and $[\mathbf{Q}_E^t]_{ij} = q(e^t = j \mid e^{t-1} = i)$, where $x$ and $e$ correspond to a generic node and edge, while the superscript indicates the time stamp.

Consequently, the transition probability at time $t$ is given by $q(\mathbf{z}^t \mid \mathbf{z}^{t-1}) = \mathbf{z}^{t-1}\mathbf{Q}^t$, where $\mathbf{z}$ stands for either $\mathbf{x}$ or $\mathbf{e}$ and the subscript of $\mathbf{Q}$ is set accordingly. Adding noise to a graph $\mathcal{G}^{t-1}$ at

time $t - 1$ results in a new graph $\mathcal{G}^t = (\mathbf{X}^t, \mathbf{E}^t)$, and involves sampling each node and edge class from a categorical distribution succinctly described as $q(\mathcal{G}^t \mid \mathcal{G}^{t-1}) = (\mathbf{X}^{t-1}\mathbf{Q}_X^t, \mathbf{E}^{t-1}\mathbf{Q}_E^t)$ and $q(\mathcal{G}^t \mid \mathcal{G}^0) = (\mathbf{X}^0\overline{\mathbf{Q}}_X^t, \mathbf{E}^0\overline{\mathbf{Q}}_E^t)$, where $\overline{\mathbf{Q}}_X^t = \mathbf{Q}_X^1 \ldots \mathbf{Q}_X^t$ and $\overline{\mathbf{Q}}_E^t = \mathbf{Q}_E^1 \ldots \mathbf{Q}_E^t$. The state transition matrices are invertible, which ensures the reversibility of the noise model: the graphs at time $t$ and time 0 can be transformed into each other using the state transition matrix. Reversibility guarantees consistency between the forward and backward processes and simplifies backward denoising.

A common issue with most known diffusion models is the large number of steps needed for convergence Hang et al. [2023], Wang et al. [2024]. For large training sets the computational complexity may be overwhelming (see our Results section). Hence, the first important question is how to reduce the number of diffusion steps for graph generation, and at the same time, preserve the chemical utility of the model. These issues are addressed by DMol.

### 3.2 DMol Forward Noise Adding Strategy

The key idea behind our approach is as follows: at each forward diffusion step $t$, we randomly select $N(t)$ nodes and $M(t)$ edges from $\mathcal{G}^0$ contained in the complete subgraph induced by the selected nodes (note that the absence of an edge is represented by a special label so that we are effectively dealing with a complete graph), and change their classes according to fixed state transition matrices $\mathbf{Q}_X$ and $\mathbf{Q}_E$, respectively, to obtain the graph $\mathcal{G}^t$. Here, $N(t)$ and $M(t)$ are **deterministic functions of the time step index** $t$. Note that we effectively run the diffusion process on a determinist number of **randomly selected subsets of nodes and edges**, with the perturbed edges confined to the subgraph induced by the selected nodes. This effectively couples the node and edge diffusion processes.

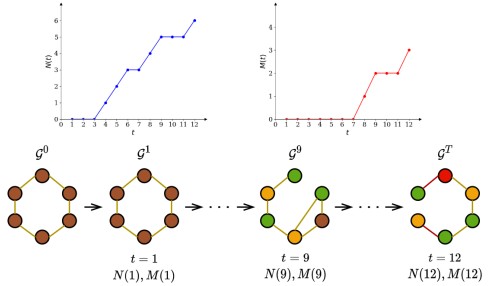
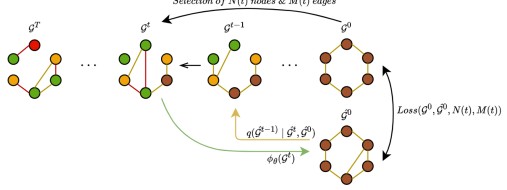

(a) At each time step $t$, we randomly select a predetermined deterministic number of nodes and add noise only to the class labels of these nodes and a fixed number of edges in the subgraph *induced* by the nodes. Both selection numbers are controlled by the time index $t$. The top row depicts how the number of selected nodes $N(t)$ and edges $M(t)$ of the selected subgraph evolves with $t$ (we set $n = 6, k = 2, r = 0.2$).

(b) DMol diffusion relies on randomly selecting $N(t)$ nodes and $M(t)$ edges from the subgraph induced by the $N(t)$ nodes at each time step $t$, and diffusing their labels to generate a noisy graph $\mathcal{G}^t$. The denoising network $\phi_\theta$ learns to predict the denoised graph from $\mathcal{G}^t$. During inference, the denoised graph is combined with $q(\hat{\mathcal{G}}^{t-1}|\hat{\mathcal{G}}^t, \hat{\mathcal{G}}^0)$ to predict $\hat{\mathcal{G}}^{t-1}$.

Figure 1: (a) The forward process; (b) DMol illustration.

For $N(t)$ and $M(t)$, we adopt cosine schedules, $N(t) = \lfloor (1 - \alpha) n \rfloor$ and $M(t) = \lfloor (1 - \alpha) N(t) (N(t) - 1) 0.5 r \rfloor$, where $\alpha = \cos^2(0.5\pi(t/T + c)/(1 + c))$, and $n$ stands for the the number of nodes in the graph, $r$ is a hyperparameter and $c$ is a small positive constant. The total number of diffusion steps is fixed at $T = k n$, where $k$ is a small hyperparameter (typically set to 1 or 2 so as to keep the number of diffusion steps as small as possible; details behind the selection of appropriate values of $k$ can be found in Appendix G.3). The maximum number of diffusion steps is proportional to the number of nodes $n$ (typically around 40 for small molecule drugs). Hence, DMol requires significantly fewer diffusion steps than other methods, such as Digress (which requires 500 or 1000 steps), and adapts itself to the size of the graph.

Additionally, our forward process constrains the class changes of edges to the subgraph induced by the nodes that underwent a state change. This ensures that the sets of altered nodes and edges are co-dependent. Still, once the selections are made, the actual state changes of nodes and edges are governed by independent processes. We next turn our attention to the choice of the transition probability matrices. The results of DiGress Vignac et al. [2023] have shown that making the

transition probability from class $i$ to class $j$ proportional to the marginal probability of class $j$ within the training dataset results in more effective learning of the true data distribution compared to when using uniform transitions. We follow the same procedure and define the marginal probabilities of node classes in the training dataset as $\mathbf{m}_x \in \mathbb{R}^{1 \times a}$ and those of edge classes as $\mathbf{m}_e \in \mathbb{R}^{1 \times b}$, respectively. Hence,

$$\mathbf{Q}_X[i,j] = \begin{cases} \frac{\mathbf{m}_x[j]}{\sum_{\ell \neq i} \mathbf{m}_x[\ell]}, & \text{if } i \neq j, \\ 0, & \text{if } i = j. \end{cases} \quad \mathbf{Q}_E[i,j] = \begin{cases} \frac{\mathbf{m}_e[j]}{\sum_{\ell \neq i} \mathbf{m}_e[\ell]}, & \text{if } i \neq j, \\ 0, & \text{if } i = j. \end{cases}$$

During the forward process, we compute the distributions of all node and edge classes after state transitions. However, we only focus on the distributions corresponding to the selected $N(t)$ nodes and $M(t)$ edges, sampling from these distributions to update their states. We preserve the states of the remaining nodes from the previous step. Algorithm 1 and Figure 1b illustrate our forward process.

---

**Algorithm 1** Forward Process

**Input:** $\mathcal{G}^0 = (\mathbf{X}^0, \mathbf{E}^0)$, state transition matrices $\mathbf{Q}_X \& \mathbf{Q}_E$, node&edge selection functions $N(\cdot) \& M(\cdot)$.
**Process:**
Sample $t \in \{1 \ldots T\}$.
Sample $\tilde{\mathcal{G}}^t = (\tilde{\mathbf{X}}^t, \tilde{\mathbf{E}}^t)$ from $\mathbf{X}^0 \mathbf{Q}_X \times \mathbf{E}^0 \mathbf{Q}_E$.
Calculate $u = N(t)$ and $v = M(t)$.
Randomly select $u$ nodes and $v$ edges from the subgraph induced by the $u$ nodes and generate "filter" matrices $\mathbf{K}_x$ and $\mathbf{K}_e$ which serve as indicators of the selections.
Set $\mathcal{G}^t = \mathcal{G}^0$ and replace $\mathbf{K}_x \mathbf{X}^t$ with $\mathbf{K}_x \tilde{\mathbf{X}}^t$, and $\mathbf{K}_e \mathbf{E}^t$ with $\mathbf{K}_e \tilde{\mathbf{E}}^t$.

---

**Algorithm 2** Sampling Process

**Input:** Denoising Network $\phi_\theta$, node&edge class distribution $\mathbf{m}_x^T = [p_i^{x,T}] \& \mathbf{m}_e^T = [p_k^{e,T}]$, state transition matrices $\mathbf{Q}_X \& \mathbf{Q}_E$.
**Process:**
Sample $\hat{\mathcal{G}}^T$ from $\mathbf{m}_x^T \times \mathbf{m}_e^T$.
**for** $t = T$ **to** 1 **do**
  $z \leftarrow f(\hat{\mathcal{G}}^t, t)$ {Structural and spectral features}.
  $\hat{p}^X, \hat{p}^E \leftarrow \phi_\theta(\hat{\mathcal{G}}^t, z, t, R_x, R_e)$ {Prediction}.
  Sample $\hat{\mathcal{G}}^0 = (\hat{\mathbf{X}}^0, \hat{\mathbf{E}}^0)$ from $\hat{p}^X \times \hat{p}^E$.
  Sample $\tilde{\mathcal{G}}^{t-1} = (\tilde{\mathbf{X}}^{t-1}, \tilde{\mathbf{E}}^{t-1})$ from $\hat{\mathbf{X}}^0 \mathbf{Q}_X \times \hat{\mathbf{E}}^0 \mathbf{Q}_E$.
  Calculate $u = N(t-1)$, $v = M(t-1)$.
  Randomly select $u$ nodes and $v$ edges from the subgraph induced by the $u$ nodes and generate corresponding "filter" matrices $\mathbf{K}_x$ and $\mathbf{K}_e$.
  Set $\hat{\mathcal{G}}^{t-1} = \hat{\mathcal{G}}^0$ and replace $\mathbf{K}_x \hat{\mathbf{X}}^{t-1}$ with $\mathbf{K}_x \tilde{\mathbf{X}}^{t-1}$, $\mathbf{K}_e \hat{\mathbf{E}}^{t-1}$ with $\mathbf{K}_e \tilde{\mathbf{E}}^{t-1}$.
**end for**

---

### 3.3 Denoising/Sampling Process

During the denoising process, we first sample a random graph based on the marginal distributions of nodes and edges at $t = T$. This random graph is then used as the input to the denoising network $\phi_\theta$ to predict a denoised graph. Based on the prediction, we use the noise model to obtain the noisy graph at $t = T - 1$. Subsequently, this noisy graph is fed into the denoising network, and the above process is repeated iteratively until a new graph is sampled.

To start the sampling process, we need to compute the marginal distributions of nodes and edges at $t = T$. At time $t = T$, a node class $i$ at $t = T$ is a result of transitions from some other node classes at $t = 0$. As a result, the marginal probability of the $i$-th node class at time $t = T$ equals

$$p_i^{x,T} = p_i^x \sum_j \frac{p_j^x}{1 - p_j^x},$$

where $p_i^x$ and $p_j^x$ are the marginal probabilities of node classes $i$ and $j$ at time $t = 0$, respectively. For edges, we similarly have that the marginal probabilities of the $i$-th edge class at $t = T$ equals

$$p_i^{e,T} = (1 - r) \, p_i^e + r \, p_i^e \sum_l \frac{p_l^e}{1 - p_l^e},$$

where $p_i^e$ and $p_l^e$ are the marginal probabilities of edge classes $i$ and $l$ at $t = 0$.

After sampling the random graph, we use the denoising model to predict the clean graph and continue this process iteratively. At time step $t$, the input to the denoising model includes not only the noisy graph and the time $t$ but also two additional functions: $R_x(t) = N(t)/n$ and

$R_e(t) = M(t)/(0.5\,n\,(n-1))$, which describe the proportion of nodes and edges in the noisy graph at time $t$ that undergo state transitions relative to the total number of nodes and edges, respectively. They indicate to the denoising network that the input noisy graph and the predicted graph should differ by $N(t)$ nodes and $M(t)$ edges, thereby helping the network make more accurate predictions. The sampling steps are listed in Algorithm 2. Figure 8 in the Appendix is an example of the sampling process from a random graph to a denoised graph. Note that the posterior distribution used by the DMol sampling process, $q(\hat{\mathcal{G}}^{t-1}|\hat{\mathcal{G}}^t, \phi_\theta)$, factorizes as $q(\hat{\mathcal{G}}^{t-1}|\hat{\mathcal{G}}^0)\,q_\theta(\hat{\mathcal{G}}^0|\hat{\mathcal{G}}^t)$. This decomposition improves both computational efficiency and inference accuracy.

The loss we use to train the denoising network differs from that of DiGress in so far that it contains two additional penalties,

$$l(\hat{\mathbf{p}}^{\mathcal{G}}, \mathcal{G}) = \sum_{1 \leq i \leq n} \mathrm{CE}(\mathbf{x}_i, \hat{\mathbf{p}}_i^X) + \lambda_1 \sum_{1 \leq i,j \leq n} \mathrm{CE}(\mathbf{e}_{ij}, \hat{\mathbf{p}}_{ij}^E)$$

$$+ \lambda_2 \mathrm{MSE}\big(D(\mathrm{argmax}(\hat{\mathbf{p}}^X); \mathrm{argmax}(\mathbf{X})), N(t)\big) + \lambda_3 \mathrm{MSE}\big(D(\mathrm{argmax}(\hat{\mathbf{p}}^E); \mathrm{argmax}(\mathbf{E})), M(t)\big).$$

where $\hat{\mathbf{p}}^{\mathcal{G}} = (\hat{\mathbf{p}}^X, \hat{\mathbf{p}}^E)$ denotes the predicted probability distributions, while $\mathbf{x}_i$ and $\mathbf{e}_{ij}$ represent the one-hot encodings of nodes and edges, respectively. CE stands for the cross-entropy loss, while MSE refers to the mean squared error loss. The term $\mathrm{argmax}(\hat{\mathbf{p}}^X)$ equals the class index of highest probability for each node, while $\mathrm{argmax}(\mathbf{X})$ effectively converts the one-hot encoding of each node into its corresponding class index. $D(\cdot; \cdot)$ is equal to the number of nodes (edges) in the two arguments that have different classes. The first two terms are the standard node classification and edge classification losses. The third and fourth terms ensure that the differences in the number of nodes and edges of the predicted and the ground truth graphs, respectively, are as close as possible to the number of nodes and edges modified during the noise addition process. This is possible since $N(t), M(t)$ are deterministic - the added losses degrade performance otherwise.

We also briefly remark that it is easy to see that the DMol diffusion model and the modified loss function are permutation invariant. To see why the second claim is true, observe that our loss function comprises two components, a cross-entropy loss and an MSE loss. For the cross-entropy loss terms,

$$l_{CE}(\hat{\mathbf{p}}^{\mathcal{G}}, \mathcal{G}) = \sum_{1 \leq i \leq n} \mathrm{CE}(\mathbf{x}_i, \hat{\mathbf{p}}_i^X) + \lambda_1 \sum_{1 \leq i,j \leq n} \mathrm{CE}(\mathbf{e}_{ij}, \hat{\mathbf{p}}_{ij}^E),$$

Since the overall cross-entropy loss is obtained by summing up the individual cross-entropy values for each node and edge, and the sum operation is permutation invariant, then the cross-entropy loss is also invariant. We arrive at a similar conclusion for the MSE terms in $l_{MSE}(\hat{\mathbf{p}}^{\mathcal{G}}, \mathcal{G})$,

$$\lambda_2 \mathrm{MSE}\big(D(\mathrm{argmax}(\hat{\mathbf{p}}^X); \mathrm{argmax}(\mathbf{X})), N(t)\big) + \lambda_3 \mathrm{MSE}\big(D(\mathrm{argmax}(\hat{\mathbf{p}}^E); \mathrm{argmax}(\mathbf{E})), M(t)\big),$$

where invariance follows based on the definition of $D(\cdot; \cdot)$ and the fact that the loss is computed separately for nodes/edges with different indices and then summed up.

### 3.4 Sampling Efficiency of DMol

To explain the sampling efficiency of DMol, we analyze the minimum time step increment $\delta t$ required to perturb exactly one node (i.e., the smallest discrete noise unit) during the diffusion process. We compare DMol with DiGress; however, it is important to note that the conclusion holds for all discrete diffusion models, including DiGress, DisCo, etc. The ratio of the required time step increments for DiGress and DMol satisfies

$$\frac{\delta t_{\mathrm{DiGress}}}{\delta t_{\mathrm{DMol}}} = \frac{1}{\sum_i p_i^x (1 - p_i^x)} > 1, \tag{1}$$

where $p_i^x$ denotes the probability of node type $i$ (for derivations, see Appendix D.2). The inequality demonstrates that DMol requires fewer steps to perturb a single node. Since the total number of diffusion steps in DMol is solely determined by the number of nodes, the efficient addition of noise to nodes directly contributes to the overall efficiency of DMol. For further discussion of likelihood computations, noise distribution evolution, and expressivity refer to the Appendices C D.

## 4 Motif Compression

To preserve motif compositions (e.g., carbon rings), we use a new motif compression feature, depicted in Figure 2. The process is designed to better preserve motifs in molecular graphs.

Table 1: Performance comparison on the QM9 dataset (see Appendix G.4 for more details).

| MODEL | V↑ | V.U.↑ | V.U.N.↑ |
|---|---|---|---|
| GRAPHNVP | 83.5 | 18.4 | - |
| GDSS | 96.0 | 94.6 | - |
| GRUM | **99.4** | 85.1 | 22.2 |
| DIGRESS | 97.8 | 95.2 | 31.8 |
| DISCO | 98.2 | 95.6 | 57.5 |
| DEFOG | 97.9 | 95.9 | 62.8 |
| DMOL(OURS) | 98.3 | **96.0** | **75.1** |

Table 2: Performance comparison on the same version of MOSES. For fair comparisons, we ran the DisCo code without any modifications, and DeFoG with 50 diffusion steps to ensure similar computational complexity to our method (DeFoG also reports their results for 50 diffusion steps).

| MODEL | V↑ | U↑ | N↑ | FILTERS↑ | FCD↓ | SNN↑ | SCAF↑ |
|---|---|---|---|---|---|---|---|
| DIGRESS | 84.8 | **100** | 94.5 | 97.2 | 1.18 | 0.55 | 14.6 |
| DISCO | 85.7 | **100** | 97.4 | 95.8 | 1.40 | 0.51 | 14.5 |
| DEFOG | 84.2 | **100** | 97.2 | 96.9 | 1.89 | 0.50 | **14.8** |
| DMOL (OURS) | **87.8** | **100** | **100** | **97.8** | **1.12** | **0.58** | **14.8** |

Specifically, we convert a small predefined number $(3 - 15)$ of most frequently occurring sub-graphs/motifs (which may differ for different training sets) that can only form single bonds with the scaffolds through their available carbons into supernodes, introducing in the process new node classes to represent the compressed motifs. The compressed graph representations are directly fed into our DMol diffusion model. During the sampling process, the supernodes are converted back to their respective motifs and integrated into the graph scaffold in a chemically valid manner.

Although superficially similar to part of the JT-VAE Jin et al. [2019] approach, motif compression is significantly different. The JT approach uses not just motifs but a large number (roughly several hundreds) of submolecular structures, and requires finding a minimum spanning tree during the encoding process; furthermore, VAEs do not directly generate graphs, and one has to perform complicated decoding. The main issue is that each compressed molecular substructure can form many different types of bonds with different atoms during the reconstruction process. The options are ranked according to a special scoring function and the overall process is computationally demanding. The motifs used in DMol for different datasets are available in the Appendix, Figures 3, 4, 5.

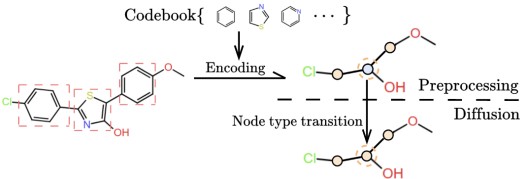

Figure 2: Motifs (i.e., substructures occurring with high frequencies or frequencies higher than predicted by random models) that also allow for unique scaffold integration are compressed into supernodes with their own labels. During diffusion, supernodes are either converted into other classes of supernodes or into atomic nodes, and vice versa. During sampling, the supernode is decoded back into its corresponding motif.

## 5 Experiments

We present next the results of running DMol on several benchmarking datasets. For experiments pertaining to other graph models (e.g., SBMs and planar graphs), the reader is referred to Appendix G.1.

**Benchmarks.** We compared DMol to several graph generation methods. The benchmarking models used in the experiments include GraphVAE Simonovsky and Komodakis [2018], GT-VAE Mitton et al. [2021], Set2GraphVAE Vignac and Frossard [2021], SPECTRE Martinkus et al. [2022b], GraphNVP Madhawa et al. [2019], GDSS Jo et al. [2022b], GruM Jo et al. [2024], DiGress Vignac et al. [2023], DisCo Xu et al. [2024b] and DeFoG Qin et al. [2025].

**Data.** We tested the performance of different generative models on the QM9 Wu et al. [2018], MOSES Polykovskiy et al. [2020], and GUACAMOL Brown et al. [2019] datasets. QM9 is a small dataset: we used 100K molecules for training, 20K for validation, and 13K for testing. Both MOSES

and GUACAMOL contain millions of molecules, with the number of heavy atoms inside a single molecule bounded by 26 and 88, respectively. We used 85% of the molecules for training, 5% for validation, and 10% for testing.

**Setup.** For each molecule dataset, we choose a different (relatively small) number of motifs to be converted into supernodes. The selected motif structures can be found in the Appendix A. We set the hyperparameters of DMol in both setting to $k = 2$, $r = 0.2$, $\lambda_1 = 5$, $\lambda_2 = 1$, $\lambda_3 = 1$. The number of training epochs is set to $500$. The experiments were conducted using NVIDIA H100 GPUs. The QM9 runs were executed on a single GPU, while the MOSES and GUACAMOL experiments were processed in parallel using 4 GPUs.

**Evaluation Metrics.** The metrics used in the experiments include validity, uniqueness, and novelty. Validity (V) measures the proportion of generated molecules that pass basic valency checks (e.g., correspond to valid SMILES files). Uniqueness (U) quantifies the proportion of generated molecules that are distinct. Novelty (N) represents the percentage of generated molecules that do not appear in the training dataset. Furthermore, V.U. stands for the product of validity and uniqueness scores, while V.U.N. stands for the product of validity, uniqueness, and novelty.

Since MOSES and GUACAMOL are benchmarking datasets, they introduce their own set of metrics for reporting results. In addition to V, U, and N, MOSES incorporates a filter score, the Frechet ChemNet Distance (FCD) Preuer et al. [2018], SNN, and scaffold similarity scores, while GUA-CAMOL uses FCD and KL divergence. The definition of these metrics are in Appendix E. The metrics mentioned above still have certain limitations. Since these metrics assess the overall quality of the generation pro-

Table 3: Classical validity, uniqueness and novelty performance comparison on GUACAMOL, for comparable generation complexity.

| MODEL | V↑ | U↑ | N↑ | KL DIV↑ | FCD↑ |
|---|---|---|---|---|---|
| DIGRESS | 84.5 | 100 | 99.8 | 93.1 | 68.4 |
| DISCO | 86.0 | 100 | 99.8 | 92.8 | 59.7 |
| DEFOG | **86.7** | 100 | 99.5 | 92.5 | 58.1 |
| DMOL (OURS) | **86.7** | 100 | **100** | **94.2** | **69.3** |

Table 4: Biochemical performance comparison on MOSES and GUACAMOL. Here, CL=ChEMBL likeness score, SD=shingle distance, ANS=average number of diffusion steps, ST=sample time, and TT=training time. Note the significantly lower ANS, ST and TT times of DMol compared to DeFoG.

| DATASET | MODEL | CL↑ | QED↑ | SD↑ | ANS↓ | ST(s)↓ | TT(H)↓ |
|---|---|---|---|---|---|---|---|
| MOSES | DIGRESS | 4.4965 | 0.8024 | 0.647 | 500 | 68.24 | 136 |
| | DISCO | 4.1373 | 0.7518 | 0.685 | 500 | 65.36 | 132 |
| | DEFOG | 4.0502 | 0.7475 | **0.693** | 50 | 5.81 | 115 |
| | DMOL (OURS) | **4.5033** | **0.8055** | 0.683 | **38** | **1.52** | **60** |
| GUACAMOL | DIGRESS | 4.0412 | 0.5650 | 0.668 | 500 | 70.82 | 162 |
| | DISCO | 3.9093 | 0.5366 | 0.679 | 500 | 68.76 | 155 |
| | DEFOG | 3.8075 | 0.4927 | **0.687** | 50 | 7.62 | 132 |
| | DMOL (OURS) | **4.2231** | **0.5786** | 0.678 | 48 | **1.84** | **88** |

cess, but do not provide insights into the performance of individual molecules. To address this limitation, we also use the ChEMBL Likeness Score (CLscore) Bühlmann and Reymond [2020], Quantitative Estimation of Drug-likeness (QED) Bickerton et al. [2012] score, and our newly introduced Shingle distance (SD) (which quantifies the divergence between two molecular motif (shingle) distributions, see the Appendix F). The CLscore is calculated by considering how many substructures in a molecule also occur in the drug-like dataset ChEMBL Gaulton et al. [2016]. QED evaluates drug-likeness by considering eight widely recognized molecular properties, such as molecular mass and polar surface area. More details are available in Appendix F. Another performance metrics used is the sample time (seconds) needed to generate a batch of $64$ molecules.

**Results.** The experimental results are presented in Tables 1, 2, 3 and 4. On QM9, DMol achieves the best performance compared to other baseline models. In contrast, GruM, DiGress, DisCo, DeFoG have low novelty scores, indicating that most of the generated molecules may be duplicates or small perturbation of those already present in the training set. On MOSES and GUACAMOL, DMol demonstrates superior performance compared to other methods with respect to all metrics.

Table 4 presents a comparison of diffusion models with respect to the CL/QED scores, average number of diffusion steps, sample time, shingle distance and training time. The results demonstrate that DMol consistently outperforms DiGress, DisCo, and DeFoG across multiple metrics. DMol produces samples with excellent chemical properties as evidenced by high CL scores and QED values. DMol also has larger SD than DiGress, indicating its ability to generate more novel molecules. Although DisCo and DeFoG achieve even higher SDs, as expected, due to lack of motif preservation, this comes at the cost of reduced chemical property preservation. Additionally, DMol exhibits excellent computational efficiency, reducing the average number of diffusion steps by an order of magnitude and significantly accelerating both sampling and training time. For a detailed ablation study, please see Appendix G.2.

# 6 Motif Distribution Analysis

To validate that motif compression does not introduce biases towards the selected compressed motifs, we conducted a comprehensive analysis comparing motif probabilities between training and generated molecular sets on the MOSES dataset. We examine the top 30 most frequent motifs, with the top 15 (IDs $0 - 14$) structures chosen for compression into supernodes, and the remaining 15 (IDs $15 - 29$) serving as a control group to assess potential distribution distortions.

**Individual Motif Probabilities.** For each motif, we compute the probability that a molecule contains that specific motif. The results (detailed in Appendix Table 10) demonstrate a close alignment between training and generated set distributions. The average supernode motif (IDs $0 - 14$) probability (IDs $0 - 14$) equals $0.1189$ in the training set and $0.1143$ in the generated set. For nonsupernode motifs (IDs 15-29), these average probabilities equal $0.0189$ and $0.0160$, respectively. This indicates that our approach does not create systematic bias between compressed and uncompressed substructures.

**Motif Co-occurrence Patterns.** We examined joint probabilities of frequent motif pairs to verify that our model maintains realistic structural relationships. For example, the co-occurrence probability of benzene (c1ccccc1) and pyridine (c1ccncc1) is $0.11$ in the training set versus $0.10$ in the generated set, and similar results hold for other pairs. This demonstrates that DMol closely preserves not only individual motif frequencies but also their co-occurrence patterns.

**Node and Supernode Marginal Distributions.** We compared the marginal distributions of all node and supernode classes of training and generated molecules (see Appendix Table 11). The distributions show strong alignment, confirming that DMol samples from the original data distribution without introducing biases. For instance, carbon atoms comprise $0.4897$ of the training atom set, while for the generated set this value equals $0.4940$; the most frequent supernode (c1ccccc1, benzene) appears with probabilities $0.0736$ and $0.0750$, respectively.

We attribute this strong preservation of substructure statistics to two key design choices: (1) using marginal distributions from the training set as priors for both supernode and atom sampling, and (2) enforcing that $N(t)$ nodes change at each timestep, ensuring adequate creation of generating nonmotif substructures. These results demonstrate that motif compression effectively preserves molecular motif distributions without introducing systematic bias.

# 7 Conditional Generation Based on Scaffolds and Docking Results

To avoid issues associated with direct conditional generation, we instead proposed a scaffold-informed pipeline. The key ideas are to cluster the training set molecules according to either their RDKit features, or embeddings generated by Mol2Vec or Grover Jaeger et al. [2018], Rong et al. [2020], and then run DMol on the samples in sufficiently large cluster groups separately (see Appendix G.6. For relevant docking results, please see Appendix I.

# 8 Conclusion

We present DMol, a highly computationally efficient motif-preserving graph diffusion for small molecule drug generation. DMol couples node and edge noise through a combination of deterministic and stochastic mechanisms, adapts the loss function and includes motif compression. On biochemically relevant performance metrics, such as the ChEMBL likeness and QED scores, it outperforms all other reported methods. **Limitations and Future Work:** Please refer to Appendix K and Appendix L.

# Acknowledgments

We gratefully acknowledge the financial support and computational resources provided by Abbvie and the NSF grant CCF 24-02815, which were essential for conducting this research. We also extend our sincere thanks to Matthew S. Krafczyk from the National Center for Supercomputing Applications (NCSA) at the University of Illinois Urbana-Champaign for his valuable assistance in resolving technical simulation issues encountered during our experiments. Furthermore, we thank Jiwoong Jung and Ziheng Qi for useful discussion.

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

# A  Motif Compression

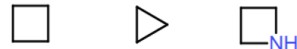

Figure 3: The 3 selected motifs for QM9.



Figure 4: The 15 selected motifs for MOSES.

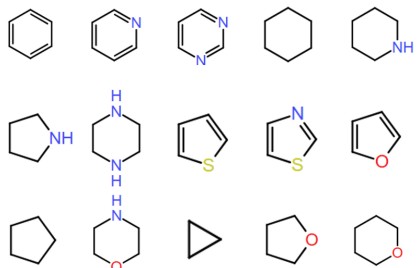

Figure 5: The 15 selected motifs for GUACAMOL.

Since each dataset has a different size of graph and complexity, we choose a different number of motifs, $K$ to be converted to supernodes. We select 3 motifs for QM9 3, 15 motifs for MOSES 4, and 15 motifs for GUACAMOL 5.

# B  DiGress vs DMol

We now further compare DiGress and DMol. First, we observe that DiGress also constructs state transition matrices based on marginal distributions, with the key difference with respect to DMol being that its transition matrices vary over time as

$$\mathbf{Q}_X^t = \alpha^t \mathbf{I} + \beta^t \mathbf{1}_a \mathbf{m}_x, \quad \mathbf{Q}_E^t = \alpha^t \mathbf{I} + \beta^t \mathbf{1}_b \mathbf{m}_e,$$

where $\alpha^t$ and $\beta^t$ are time-dependent scaling parameters, while $\mathbf{Q}_X^t$ and $\mathbf{Q}_E^t$ are the state transition matrices used to transition from $\mathcal{G}^{t-1}$ to $\mathcal{G}^t$. However, since the forward noise adding process is Markovian, DiGress directly utilizes $\overline{\mathbf{Q}}_X^t$ and $\overline{\mathbf{Q}}_E^t$ to achieve the transition from $\mathcal{G}^0$ to $\mathcal{G}^t$, where $\overline{\mathbf{Q}}_X^t = \mathbf{Q}_X^1 \ldots \mathbf{Q}_X^t$ and $\overline{\mathbf{Q}}_E^t = \mathbf{Q}_E^1 \ldots \mathbf{Q}_E^t$.

Since $(\mathbf{1m})^2 = \mathbf{1m}$, we have

$$\overline{\mathbf{Q}}_X^t = \overline{\alpha}^t \mathbf{I} + \overline{\beta}^t \mathbf{1}_a \mathbf{m}_x, \overline{\mathbf{Q}}_E^t = \overline{\alpha}^t \mathbf{I} + \overline{\beta}^t \mathbf{1}_b \mathbf{m}_e,$$

where $\overline{\alpha}^t = \prod_{\tau=1}^t \alpha^\tau$ and $\overline{\beta}^t = 1 - \overline{\alpha}^t$. Additionally, DiGress also adopts a cosine schedule for $\overline{\alpha}^t$: $\overline{\alpha}^t = \cos^2(0.5\pi(t/T + c)/(1 + c))$, where $c > 0$ is a constant.

In DMol, for a graph dataset with $n$ nodes, the number of nodes undergoing state changes at time step $t$ equals $N(t) = (1 - \alpha)\, n$. At the same time step in Digress, the expected number of nodes undergoing state changes equals $\sum_i n p_i^x (1 - \alpha)(1 - p_i^x)$, where $p_i^x$ represents the marginal probability of node class $i$. Therefore, at the same time step, the number of nodes modified by DMol is $\frac{1}{\sum_i p_i^x (1 - p_i^x)}$ times that expected from Digress (the derivations can be found in Appendix D.2). Moreover, among the nodes whose states are changed by DMol, the proportion of nodes with class $i$ is $p_i^x$, whereas in DiGress, this proportion equals $p_i^x (1 - p_i^x)$. The ratio of these fractions equals $1 - p_i^x$.

For edges, a similar analysis can be applied, leading to the conclusion that at the last time step $T$, the ratio of the number of edges modified by our noise model and that modified by Digress equals $\frac{r}{\sum_j p_j^e (1 - p_j^e)}$, where $r$ is the hyperparameter mentioned in Section 3.2. At each time step, this ratio needs to be adjusted by a correction factor that accounts for the fact that our method only selects edges from the complete graph induced by the chosen nodes. This correction factor is the ratio of the number of edges in the induced subgraph and the total number of edges. Additionally, among the modified edges, the proportion of edges with class index $j$ differs from the same in DiGress by a constant factor of $1 - p_j^e$, where $p_j^e$ equals the marginal probability of edge class $j$. As a result, DMol can be made to replicate the average "behavior" of DiGress by simply adjusting specific hyperparameters, whereas the opposite claim is not true. Comparisons of the noise evolution distribution, efficiency and expressivity of DMol and Digress are available in Appendix D.

The following modifications can be applied to DMol to obtain Digress:

1. Fix the maximum number of sampling steps to a constant length instead of varying based on the number of nodes.

2. Scale the number of nodes to be changed at each step by a factor of $\sum_i p_i^x (1 - p_i^x)$.

3. Set the noising processes for edges and nodes to be independent, i.e., select edges from the entire graph instead of restricting to the subgraph formed by the selected nodes.

4. Scale the number of edges to be changed at each step by a factor of $\frac{\sum_j p_j^e (1 - p_j^e)}{r}$.

5. When generating random scores for nodes and edges to determine the objects to be selected, multiply the random scores for nodes by a correction weight of $1 - p_i^x$, and multiply the random scores for edges by $1 - p_j^e$.

These modifications are straightforward to implement in practice and can be achieved by simply adjusting model hyperparameters. However, Digress cannot be easily converted into DMol.

## C  Likelihood Computation

DMol can be visualized through the model depicted in Figure 6, informed by the dataset distribution and where nodes are perturbed first, while edges are conditionally perturbed based on the node selection. This framework extends traditional hierarchical variational inference structures by incorporating a crucial dependency relationship: the selection of edges to perturb is conditioned on the selection of nodes at each timestep.

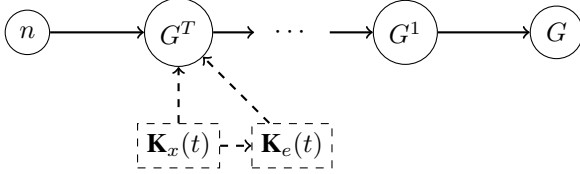

Figure 6: The conditional noise-addition model of DMol, where $\mathbf{K}_x(t)$ captures the random node selection while $\mathbf{K}_e(t)$ captures the random edge selection process; the dashed arrow indicating the dependency of edge selection on node selection.

For a molecular graph $G$, the model likelihood must account for this conditional relationship, according to:

$$\log p_\theta(G) = \log \sum_{n \in \mathcal{N}} p(n) \int p(G^T|n) p_\theta(G^{T-1}, \ldots, G^1|G^T) p_\theta(G|G^1) d(G^1, \ldots, G^T) \tag{2}$$

$$= \log p(n_G) + \log \int p(G^T|n_G) \prod_{t=2}^T p_\theta(G^{t-1}|G^t, \mathbf{K}_x(t), \mathbf{K}_e(t)) p_\theta(G|G^1) d(G^1, \ldots, G^T) \tag{3}$$

Note the key difference in the conditional probability $p_\theta(G^{t-1}|G_t, \mathbf{K}_x(t), \mathbf{K}_e(t))$, which explicitly accounts for the selection matrices $\mathbf{K}_x(t)$ and $\mathbf{K}_e(t)$ that designate which nodes and edges receive perturbations at time $t$.

Following established variational inference principles, we derive a tractable evidence lower bound (ELBO) adapted for this conditional structure:

$$\log p_\theta(G) \geq \log p(n_G) + \underbrace{D_{KL}[q(G^T|G)\|q_\mathcal{X}(n_G) \times q_\mathcal{E}(n_G)]}_{\text{Prior term}} \tag{4}$$

$$+ \underbrace{\sum_{t=2}^T \mathcal{L}_t}_{\text{Diffusion component}} \tag{5}$$

$$+ \underbrace{E_{q(G^1|G)}[\log p_\theta(G|G^1)]}_{\text{Reconstruction component}} \tag{6}$$

Where the diffusion component $\mathcal{L}_t$ now explicitly models the conditional relationship between node and edge noise:

$$\mathcal{L}_t(G) = E_{q(G^t|G)} \left[ D_{KL} \left[ q(G^{t-1}|G^t, G, \mathbf{K}_x(t-1), \mathbf{K}_e(t-1)) \| p_\theta(G^{t-1}|G^t, \mathbf{K}_x(t-1), \mathbf{K}_e(t-1)) \right] \right] \tag{7}$$

This can be further decomposed to highlight the conditional structure:

$$\mathcal{L}_t = \sum_{i \in K_x(t-1)} E_{q(x_i^t|x_i)} \left[ D_{KL} \left[ q(x_i^{t-1}|x_i^t, x_i) \| p_\theta(x_i^{t-1}|G^t) \right] \right] \tag{8}$$

$$+ \sum_{(i,j) \in K_e(t-1)} E_{q(e_{ij}^t|e_{ij})} \left[ D_{KL} \left[ q(e_{ij}^{t-1}|e_{ij}^t, e_{ij}) \| p_\theta(e_{ij}^{t-1}|G^t, \mathbf{K}_x(t-1)) \right] \right] \tag{9}$$

Note the critical difference in the edge term, where the predicted probability $p_\theta(e_{ij}^{t-1}|G^t, \mathbf{K}_x(t-1))$ is explicitly conditioned on the node selection $\mathbf{K}_x(t-1)$. This formulation captures the key aspect of DMol's noise process: edges are selected for perturbation only from within the subgraph induced by the selected nodes.

The conditional dependency structure provides two key advantages for DMol: 1. It preserves local structural coherence (e.g, motifs) by ensuring that edge modifications are correlated with node changes. 2. It significantly reduces the computational complexity by restricting edge perturbations to smaller subgraphs.

In practice, each ELBO component remains computationally tractable: we calculate $\log p(n_G)$ from the empirical node count distributions across the molecular corpus. The prior term involves categorical KL divergence calculations, while the diffusion component now incorporates the conditional selection of edges based on the node selection. The reconstruction component derives from generated probabilities of the original structure given its slightly perturbed version $G^1$.

# D    Theoretical Analysis of DMol

## D.1    Noise Distribution Evolution

In DiGress, the node type distribution remains fixed at all forward time steps because the initial node type distribution is the marginal distribution $\mathbf{m}_X$. This is easy to see because $p_0 = \mathbf{m}_X$, so $p_1 = \overline{\alpha}^1 p_0 + \overline{\beta}^1 \mathbf{m}_X = \mathbf{m}_X$, and by induction, $p_t = \mathbf{m}_X$ for all $t$, where $p_t$ is the node type distribution at step $t$. In DMol, the node type distribution changes over time. For a node with type $i$ selected for perturbation, the probability of transitioning to type $j \neq i$ is $P(j|i_{\text{selected}}) = \frac{p_X(j)}{1 - p_X(i)}$. This leads to a different recurrence relation for the probability distribution over time.

## D.2    Efficiency

To compare the efficiency of DMol and DiGress, we compare the time step increment required to change exactly one node(minimal noise) under both settings. In DMol, the number of nodes changing at time $t$ is $N(t) = (1 - \alpha^t) \cdot n$. To find $\delta t_{DMol}$ such that $\delta N(t) = 1$, we need
$$N(t + \delta t) - N(t) = 1$$
The number of nodes that change at time $t + \delta t$ equals
$$N(t + \delta t) = (1 - \alpha(t + \delta t)) \cdot n$$
For DMol, we use $T = kn$ and the cosine schedule for $\alpha$,
$$\alpha(t) = \cos^2 \left( \frac{0.5\pi(t/T + s)}{1 + s} \right).$$
Taking the derivative with respect to $t$ and using a small angle approximation for small $\delta t$, we obtain:
$$\frac{d\alpha(t)}{dt} \approx -\frac{\pi \sin\left(\frac{\pi(t/T + s)}{1+s}\right) \cos\left(\frac{\pi(t/T + s)}{1+s}\right)}{T(1 + s)} = -\frac{\pi \sin\left(\frac{2\pi(t/T + s)}{1+s}\right)}{2T(1 + s)}$$
For small $\delta t$, we can approximate:
$$\alpha(t + \delta t) - \alpha(t) \approx \frac{d\alpha(t)}{dt} \cdot \delta t$$
Therefore,
$$N(t + \delta t) - N(t) = (\alpha(t) - \alpha(t + \delta t)) \cdot n \approx -\frac{d\alpha(t)}{dt} \cdot \delta t \cdot n$$
Setting this equal to 1 and solving for $\delta t$, we can have the value of $\delta t_{DMol}$:
$$\delta t_{DMol} = \frac{1}{-\frac{d\alpha(t)}{dt} \cdot n} = \frac{2T(1 + s)}{\pi n \sin\left(\frac{2\pi(t/T + s)}{1+s}\right)}$$

For DiGress, we use a similar approach to find $\delta t_{Digress}$. First, the expected number of nodes changed by DiGress at time $t$ equals:
$$N_{Digress}(t) = n \cdot (1 - \alpha(t)) \cdot \sum_i p_i^x \cdot (1 - p_i^x)$$
Following a similar derivation as for DMol, we arrive at
$$\delta t_{Digress} = \frac{1}{-\frac{d\alpha(t)}{dt} \cdot n \cdot \sum_i p_i^x \cdot (1 - p_i^x)} = \frac{2T(1 + s)}{\pi n \sin\left(\frac{2\pi(t/T + s)}{1+s}\right) \cdot \sum_i p_i^x \cdot (1 - p_i^x)}$$
Then, the ratio of the minimum time steps of interest equals
$$\frac{\delta t_{Digress}}{\delta t_{DMol}} = \frac{1}{\sum_i p_i^x \cdot (1 - p_i^x)}$$
Because $\sum_i p_i^x \cdot (1 - p_i^x) < 1$ for any probability distribution, we have that
$$\frac{\delta t_{Digress}}{\delta t_{DMol}} > 1$$

This proves that $\delta t_{Digress} > \delta t_{DMol}$, or, in words, that DMol requires a smaller time step increment to change exactly one node. As a result, the number of steps required by DMol to change a node will be smaller than that of required by Digress, which implies that DMol performs graph perturbations more efficiently.

### D.3 Expressivity

Both DiGress and DMol start with the same underlying node and edge type distributions, but differ in how efficiently they explore the space of possible graphs. DMol tends to produce more diverse graphs within its reachable set because it makes larger localized changes. The Hamming distance between the initial graph $\mathcal{G}_0$ and the graph at time $t$, $\mathcal{G}_t$, is approximately $d_H(\mathcal{G}_0, \mathcal{G}_t) \sim \beta^t \cdot n$. This Hamming distance captures the difference between the number of node/edge types in the two graphs. For DiGress, the expected Hamming distance is $\mathbb{E}[d_H(\mathcal{G}_0, \mathcal{G}_t)] \sim \beta^t \cdot n \cdot \sum_{i \in \mathcal{C}_X} p_X(i) \cdot (1 - p_X(i))$. Since $\sum_{i \in \mathcal{C}_X} p_X(i) \cdot (1 - p_X(i)) < 1$, we conclude that DMol explores the graph space more efficiently per diffusion step.

## E  The Metrics used in MOSES and GUACAMOL

The metrics used in MOSES and GUACAMOL dataset are defined as follows: The filter score measures the proportion of molecules that pass the same filters used to construct the test set. The FCD score measures the similarity between molecules in the training and test sets using embeddings learned by a neural network. SNN quantifies the similarity of a molecule to its nearest neighbor based on the Tanimoto distance. Scaffold similarity compares the frequency distributions of so-called Bemis-Murcko scaffolds. KL divergence measures the differences in the distributions of various physicochemical descriptors.

## F  ChEMB Likeness, QED Scores, and Shingle Distance

The ChEMBL-Likeness Score (CLscore) Bühlmann and Reymond [2020] is a metric used to assess the drug-likeness of a molecule. It is determined by analyzing how many of the molecule's substructures are present in the ChEMBL datasetGaulton et al. [2016], a database of bioactive molecules with drug-like properties. Essentially, the CLscore quantifies the likelihood of a molecule being drug-like based on its substructure similarity to known compounds in ChEMBL.

Quantitative Estimation of Drug-likeness (QED) Bickerton et al. [2012] is a computational metric used in drug discovery to assess how "drug-like" a compound is. It combines eight physicochemical properties, including molecular mass ($M_r$), octanol–water partition coefficient (ALOGP), number of hydrogen bond donors (HBDs), number of hydrogen bond acceptors (HBAs), molecular polar surface area (PSA), number of rotatable bonds (ROTBs), number of aromatic rings (AROMs), and number of structural alerts (ALERTS), into a single score ranging from 0 (least drug-like) to 1 (most drug-like). The score is measured through defined desirability functions of the eight properties. QED provides a balanced evaluation by weighting these properties based on their distributions in known drugs.

The shingles distance (SD) is a metric designed to quantify the structural divergence between generated molecules and a reference dataset. It is computed by comparing the frequencies of important, predefined molecular drug substructures (motifs), known in RdKit as shingles, according to the ChEMBL dataset Gaulton et al. [2016]. The same set of shingles is used in the computation of the ChEMBL-Likeness Score (CLscore) Bühlmann and Reymond [2020]. SD measures the cosine distance between the vectors of shingle occurrence counts for two molecular distributions. By emphasizing differences in the presence of key drug-relevant motifs, SD offers a more nuanced assessment of both validity and novelty.

## G  Supplementary Experiments

### G.1  General Graph Generation

We conducted experiments using the benchmark proposed by Martinkus et al. [2022a], which comprises two datasets: SBM and Planar. Each dataset consists of 200 graphs. DMol's ability to accurately model various properties of these graphs was assessed using metrics such as the degree distribution (Deg.), clustering coefficient distribution (Clus.), and orbit count distribution (Orb., the number of occurrences of substructures with 4 nodes), measured by the relative squared Maximum Mean Discrepancy (MMD).

Table 5: MMD Performance Comparison on SBM and Planar Graphs.

| DATASET | MODEL | DEG ↓ | CLUS ↓ | ORB ↓ | VALID ↑ |
|---------|-------|-------|--------|-------|---------|
| SBM | GRAPHRNN | 6.7 | 1.6 | 3.2 | 5.2 |
| | GRAN | 14.3 | 1.7 | 2.0 | 25.5 |
| | SPECTRE | 1.9 | 1.7 | 1.5 | 100.0 |
| | DIGRESS | 1.7 | 1.6 | 1.7 | 66.7 |
| | DMOL(OURS) | 1.6 | 1.5 | 1.5 | 66.1 |
| PLANAR | GRAPHRNN | 24.6 | 8.8 | 2534.0 | 0.0 |
| | GRAN | 3.6 | 1.2 | 1.8 | 98.2 |
| | SPECTRE | 2.6 | 2.5 | 2.5 | 100.0 |
| | DIGRESS | 1.6 | 1.5 | 1.6 | 84.5 |
| | DMOL(OURS) | 1.8 | 1.6 | 1.4 | 83.9 |

Table 6: Ablation Study

| METHOD | V ↑ | U ↑ | N ↑ |
|--------|-----|-----|-----|
| ONLY CE LOSS | 79.6 | 100 | 100 |
| BOTH CE & MSE LOSS | 85.4 | 100 | 100 |
| SELECTING EDGES FROM THE WHOLE GRAPH | 55.6 | 100 | 100 |
| SELECTING EDGES FROM THE INDUCED SUBGRAPH | 85.4 | 100 | 100 |

MMD measures the discrepancy between two sets of distributions. The relative squared MMD is defined as follows:

$$\frac{\text{MMD}^2\left(\mathcal{G}_{gen}\|\mathcal{G}_{test}\right)}{\text{MMD}^2\left(\mathcal{G}_{train}\|\mathcal{G}_{test}\right)},$$

where $\mathcal{G}_{gen}$, $\mathcal{G}_{train}$, and $\mathcal{G}_{test}$ are the sets of generated graphs, training graphs, and test graphs.

In our experiments, we split the datasets into training, validation, and test sets with proportions of 64%, 16%, and 20%, respectively. The chosen hyperparameters were $k = 2$ and $r = 0.01$. The experimental results are presented in Table 5. These results demonstrate that DMol performs comparably to DiGress in generating general graphs.

## G.2 Ablation Study

The ablation study highlights the performance improvements of DMol compared to DiGress due to the use of a new loss function that penalizes discrepancies in the counts of nodes and edges of training molecular graphs and sampled graphs; and performance improvements due to the use of special subgraph selection procedures. The results are summarized in Table 6. The model we used is DMol, and the dataset is GUACAMOL. In Table 6, "ONLY CE LOSS" stands for using only the cross-entropy loss (the loss used by DiGress), while "BOTH CE & MSE LOSS" refers to using both cross-entropy loss and mean squared error loss (the loss used by DMol). "SELECTING EDGES FROM THE WHOLE GRAPH" means selecting edges from the entire graph, whereas "SELECTING EDGES FROM THE INDUCED SUBGRAPH" means selecting edges only from the induced subgraph formed by the selected nodes. The results show that incorporating the MSE loss can improve the validity of the generated molecules. This is because the MSE loss penalizes discrepancies between the number of altered nodes and edges and the noise added during the forward process, resulting in more accurate predictions by the denoising model. Additionally, selecting edges from the induced subgraph significantly enhances the validity of the generated molecular graphs. This is achieved by making the sets of altered nodes and edges codependent. However, if edges are selected from the whole graph, the reduced number of diffusion steps significantly increases the learning difficulty for the denoising model, leading to lower validity in the generated molecular graphs.

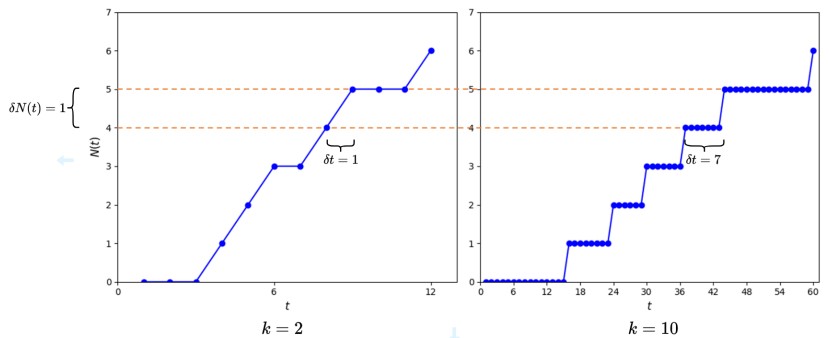

Figure 7: Choosing a larger value of $k$ results in a less step-like (flatter) $N(t)$ function. As shown in the two figures, using $k = 10$ leads to a more skewed distribution of $\delta t$ toward $N(t)$. This means that during sampling, the model is more likely to select $t$ during the early diffusion stages, which makes it harder for the neural network model to learn how to denoise.

Table 7: DMol with different values of $k$ on MOSES dataset

| $k$ | VALIDITY ↑ | UNIQUENESS ↑ | NOVELTY ↑ | SAMPLE TIME ↓ |
|-----|------------|--------------|-----------|---------------|
| 1 | 85.1 | 100 | 100 | 2.53 |
| 2 | **85.6** | 100 | 100 | 5.28 |
| 3 | 84.9 | 100 | 100 | 7.81 |
| 4 | 84.7 | 100 | 100 | 10.37 |
| 5 | 84.4 | 100 | 100 | 12.69 |
| 10 | 83.4 | 100 | 100 | 24.76 |

### G.3   Selection of the parameter $k$

The hyperparameter $k$ determines the total number of diffusion steps since we set $T = kn$, where $n$ is the number of nodes. In practice, $k$ is typically set to 1 or 2 to reduce the number of diffusion steps and enhance performance (as explained in Figure 7). The latter observation can be intuitively explained as follows: the number of nodes modified at each time step increases over time, so that by the final time step, all nodes are changed. As $k$ increases, the x-axis of the diffusion process is effectively stretched by a factor of $k$, while the y-axis remains unchanged (Figure 7). Consequently, the number of diffusion steps required to modify a single node increases. This leads to a higher frequency of generated samples during the early stages of diffusion, when fewer nodes are altered; furthermore, fewer samples are generated during the "middle stages." However, these additional early-stage samples introduce redundancy during the process of training the denoising network, ultimately degrading performance. Furthermore, these samples are generated at the cost of samples with a larger number of node changes, further degrading the performance.

In addition, to empirically evaluate the impact of $k$ on the performance of the diffusion model, we conducted experiments for different values of $k$. As shown in Table 7, the sample time is approximately proportional to $k$. Moreover, setting $k = 2$ yields the highest validity.

### G.4   Complete Experimental Results for the QM9 Dataset

For complete performance comparison on the QM9 dataset, please refer to Table 8.

### G.5   Forward Process Schedule

To examine the impact of the cosine schedule on different molecular scaffolds and topologies, we conducted an ablation study comparing different values of the hyperparameter $c$ in the cosine schedule $\alpha = \cos^2(0.5\pi(t/T+c)/(1+c))$, as well as comparing cosine versus linear schedules on the MOSES dataset. The results are presented in Table 9.

Table 8: Performance comparison on the QM9 dataset.

| MODEL | V↑ | V.U.↑ | V.U.N.↑ |
|---|---|---|---|
| GRAPHVAE | 56.1 | 42.8 | 26.5 |
| GT-VAE | 74.8 | 16.5 | 15.6 |
| SET2GRAPHVAE | 60.0 | 56.8 | - |
| SPECTRE | 87.5 | 31.4 | 29.0 |
| GRAPHNVP | 83.5 | 18.4 | - |
| GDSS | 96.0 | 94.6 | - |
| GRuM | **99.4** | 85.1 | 22.2 |
| DIGRESS | 97.8 | 95.2 | 31.8 |
| DISCO | 98.2 | 95.6 | 57.5 |
| DEFOG | 97.9 | 95.9 | 62.8 |
| DMOL(OURS) | 98.3 | **96.0** | **75.1** |

Table 9: Ablation study for forward process schedules on the MOSES dataset. The cosine schedule with $c = 0.008$ achieves optimal performance across all metrics.

| Method | Validity↑ | Uniqueness↑ | Novelty↑ | Filters↑ | FCD↓ | SNN↑ | Scaf↑ | ChEMBL↑ | QED↑ |
|---|---|---|---|---|---|---|---|---|---|
| cos ($c = 0.004$) | 86.9 | 100 | 100 | 97.4 | 1.13 | 0.57 | 14.2 | 4.5022 | 0.8022 |
| cos ($c = 0.006$) | 87.5 | 100 | 100 | 97.7 | 1.12 | 0.57 | 14.6 | 4.5020 | 0.8037 |
| cos ($c = 0.008$) | **87.8** | **100** | **100** | **97.8** | **1.12** | **0.58** | **14.8** | **4.5033** | **0.8055** |
| cos ($c = 0.01$) | 87.3 | 100 | 100 | 97.0 | 1.19 | 0.55 | 14.3 | 4.5018 | 0.8042 |
| Linear | 85.2 | 99.8 | 99.9 | 96.1 | 1.19 | 0.56 | 14.2 | 4.5011 | 0.8016 |

The experimental results demonstrate that varying the value of $c$ has a relatively small impact on crucial performance metrics, with an optimal performance achieved at $c = 0.008$, which is the setting used in our experiments. This finding aligns with the findings by DiGress, which also adopts $c = 0.008$ as the default value.

When comparing cosine and linear schedules, we see that the cosine schedule clearly outperforms the linear schedule. This advantage stems from the fact that the cosine schedule adds less noise per timestep during both the initial and final phases of the diffusion process, compared to intermediate stages. This distribution better aligns with the learning characteristics of diffusion models. Specifically, maintaining relatively small noise levels during the early and final diffusion phases facilitates the model's learning of fine-grained data features and ensures "smoother" convergence. In contrast, the uniform noise distribution in linear schedules may prove either overly aggressive or overly conservative at certain stages, leading to compromised training efficiency and performance.

### G.6 Scaffold-Constrained and Conditional Generation

To avoid issues associated with direct conditional generation, we instead proposed a scaffold-informed pipeline. The key ideas are to cluster the training set molecules according to either their RDKit features, or embeddings generated by Mol2Vec or Grover Jaeger et al. [2018], Rong et al. [2020], and then run DMol on the samples in sufficiently large cluster groups separately. More details are provided below.

**RDKit-Based Scaffold Classification:** Molecules are categorized into four scaffold classes following RDKit recommendations: aromatic monocyclic, aromatic monocyclic heterocycle, fused bicyclic, and aromatic heterocyclic.

**GNN, Transformer and Other Embedding Classification Methods:** One can use Mol2Vec or Grover to perform molecular graph embeddings, and follow up with Kmeans++ clustering.

The detailed analysis of the pros and cons of this approach compared to classical conditional generation will be discussed in more detail in a companion paper.

### G.7 Motif Probability Distribution Analysis

Table 10 presents the complete per-motif probabilities for the top 30 most frequent motifs in the MOSES dataset. Table 11 shows the marginal distributions of node and supernode categories.

Table 10: Comparison of motif probabilities between training and generated sets on the MOSES dataset. IDs $0 - 14$ correspond to supernodes, while IDs $15 - 29$ are nonsupernode motifs.

| ID | Motif (SMILES) | Prob on Training Set | Prob on Generated Set |
|----|----------------|---------------------|----------------------|
| *Supernode Motifs (IDs $0 - 14$)* | | | |
| 0 | c1ccccc1 | 0.7753 | 0.7861 |
| 1 | c1ccncc1 | 0.1829 | 0.1630 |
| 2 | c1cnnc1 | 0.1019 | 0.0912 |
| 3 | C1CCNCC1 | 0.0852 | 0.0783 |
| 4 | C1CCNC1 | 0.0798 | 0.0662 |
| 5 | c1cscc1 | 0.0763 | 0.0740 |
| 6 | c1ccsn1 | 0.0715 | 0.0686 |
| 7 | C1COCCN1 | 0.0625 | 0.0651 |
| 8 | C1CNCCN1 | 0.0620 | 0.0681 |
| 9 | c1ccoc1 | 0.0573 | 0.0429 |
| 10 | c1cncnc1 | 0.0559 | 0.0512 |
| 11 | c1cncn1 | 0.0464 | 0.0416 |
| 12 | c1ncon1 | 0.0459 | 0.0458 |
| 13 | c1ncnn1 | 0.0453 | 0.0415 |
| 14 | C1CCCCC1 | 0.0351 | 0.0301 |
| *Nonsupernode Motifs (IDs 15-29)* | | | |
| 15 | C1CC1 | 0.0298 | 0.0242 |
| 16 | c1cnoc1 | 0.0289 | 0.0238 |
| 17 | C1CCCC1 | 0.0246 | 0.0281 |
| 18 | C1CCOC1 | 0.0231 | 0.0241 |
| 19 | c1ccnc1 | 0.0214 | 0.0191 |
| 20 | c1cCNCC1 | 0.0202 | 0.0169 |
| 21 | c1ccnnc1 | 0.0191 | 0.0123 |
| 22 | c1nnnn1 | 0.0177 | 0.0161 |
| 23 | c1nncs1 | 0.0164 | 0.0143 |
| 24 | c1cnccn1 | 0.0154 | 0.0127 |
| 25 | c1cnnn1 | 0.0151 | 0.0104 |
| 26 | c1cocn1 | 0.0146 | 0.0088 |
| 27 | c1nnco1 | 0.0135 | 0.0090 |
| 28 | c1cCCCC1 | 0.0123 | 0.0098 |
| 29 | c1cNCC1 | 0.0114 | 0.0102 |
| *Average Probabilities* | | | |
| Top 15 (Supernodes) mean | | 0.1189 | 0.1143 |
| Last 15 (Nonsupernodes) mean | | 0.0189 | 0.0160 |

# H Examples of Generated Molecules.

Figure 8 illustrates the sampling process.

Figure 9, 10, 11 shows examples of generated molecules.

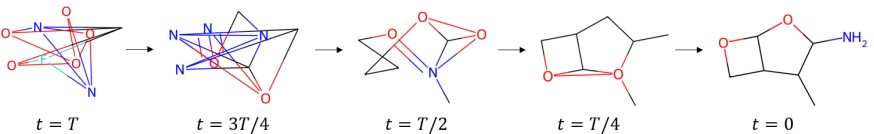

Figure 8: Sampling process.

Table 11: Marginal distributions of node and supernode categories in the MOSES dataset. Supernodes are treated as new node types alongside individual atoms.

| Node / Supernode Type | Training Set | Generated Set |
|---|---|---|
| *Individual Atoms* | | |
| C | 0.4897 | 0.4940 |
| N | 0.1511 | 0.1506 |
| S | 0.0170 | 0.0167 |
| O | 0.1654 | 0.1644 |
| F | 0.0246 | 0.0243 |
| Cl | 0.0093 | 0.0091 |
| Br | 0.0025 | 0.0026 |
| *Supernode Motifs* | | |
| c1ccccc1 | 0.0736 | 0.0750 |
| c1ccncc1 | 0.0132 | 0.0130 |
| c1cnnc1 | 0.0081 | 0.0077 |
| C1CCNCC1 | 0.0068 | 0.0070 |
| C1CCNC1 | 0.0049 | 0.0045 |
| c1cscc1 | 0.0055 | 0.0052 |
| c1ccsn1 | 0.0056 | 0.0050 |
| C1COCCN1 | 0.0035 | 0.0036 |
| C1CNCCN1 | 0.0036 | 0.0031 |
| c1ccoc1 | 0.0041 | 0.0039 |
| c1cncnc1 | 0.0044 | 0.0040 |
| c1cncn1 | 0.0010 | 0.0008 |
| c1ncon1 | 0.0027 | 0.0025 |
| c1ncnn1 | 0.0024 | 0.0020 |
| C1CCCCC1 | 0.0010 | 0.0008 |

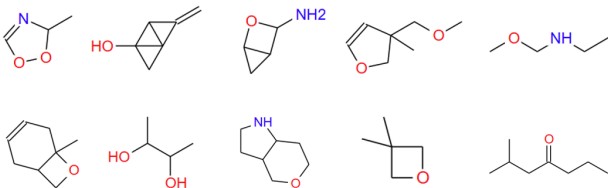

Figure 9: Molecular graphs sampled from DMol trained on the QM9 dataset.

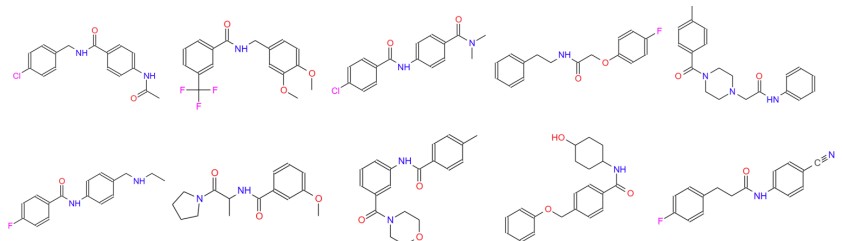

Figure 10: Molecular graphs sampled from DMol trained on the MOSES dataset.

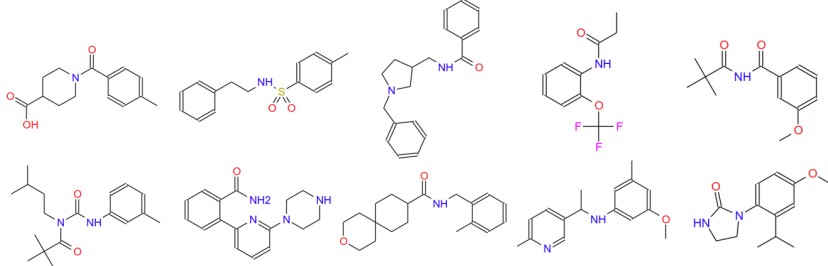

Figure 11: Molecular graphs sampled from DMol trained on the GUACAMOL dataset.

# I Sample Docking Results

We used AutoDock Vina version 1.2.0 Goodsell and Olson [1990] to visualize the docking of two of the top-scoring generated molecules (with respect to the ChEMBL likeness) that also exhibit strong biding affinities on the 1PXH and 4MQS proteins when compared to reference drug ligants (the reason behind the selection of these proteins is confidential information). The 4MQS complex represents the active human M2 muscarinic acetylcholine receptor bound to the antagonist iperoxo, while 1PXH represents the protein tyrosine phosphatase 1B with bidentate inhibitor compound 2. The free energy results are shown in Figures 12 and 13, and they indicate that the generated molecules can significantly outperform or underperform existing reference drugs.

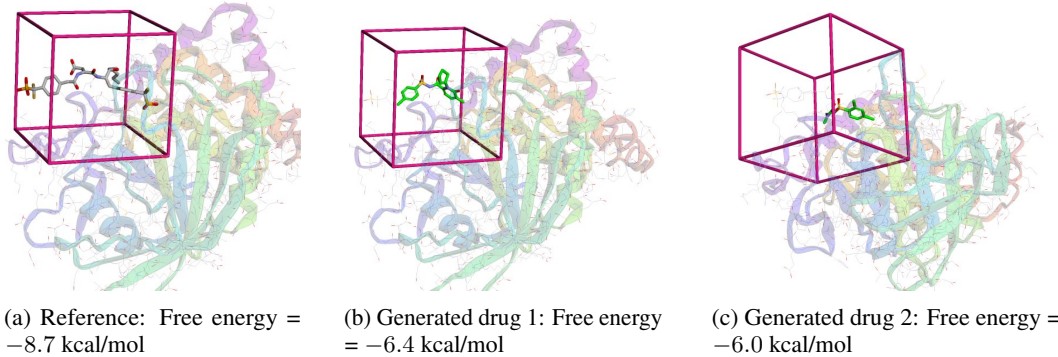

(a) Reference: Free energy = −8.7 kcal/mol

(b) Generated drug 1: Free energy = −6.4 kcal/mol

(c) Generated drug 2: Free energy = −6.0 kcal/mol

Figure 12: The reference molecule (a) and two of our generated molecules (b) and (c) docked at the protein receptor 1PXH, and the corresponding free energies. The generated molecules significantly underperform with respect to the reference due to their smaller sizes/masses. As a result, molecular weight constraints have to be considered as part of the design process.

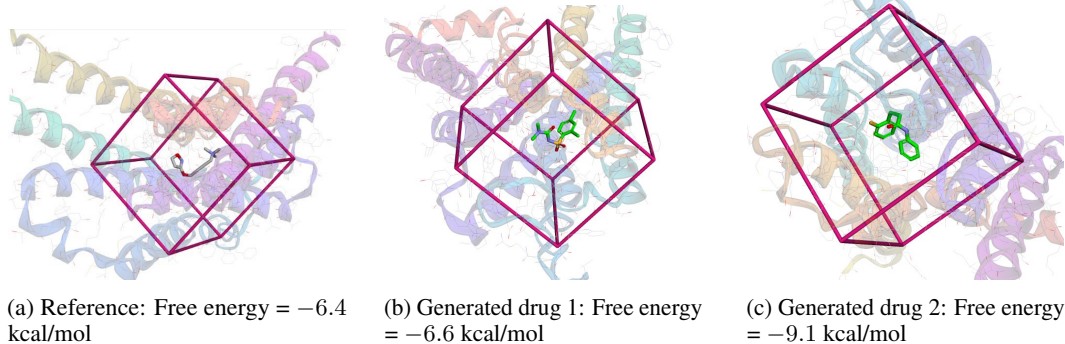

(a) Reference: Free energy = −6.4 kcal/mol

(b) Generated drug 1: Free energy = −6.6 kcal/mol

(c) Generated drug 2: Free energy = −9.1 kcal/mol

Figure 13: The reference molecule (a) and two of our generated molecules (b) and (c) docked at the protein receptor 4MQS, and the corresponding free energies. The generated molecules significantly outperform with respect to the reference, and the sizes/masses of the drug molecules are comparable. This confirms the importance of focusing on the right molecular weight/size.

## J  Broader Impact

This work presents DMol, an efficient molecular graph generation framework with a strong potential for practical applications in accelerating drug discovery, designing novel materials, and advancing computational chemistry. By substantially reducing computational requirements while maintaining or improving molecular quality metrics, our approach could democratize access to molecular design tools, enabling researchers with limited computational resources to engage in this field. The efficiency gains may also enable exploration of larger chemical spaces, potentially leading to discoveries that address pressing challenges in healthcare, energy storage, and environmental remediation. However, we acknowledge that molecular generation technologies could potentially be misused for designing harmful substances if deployed without appropriate safeguards. Additionally, as with many other AI systems, there is risk of amplifying biases present in training data, which could limit the diversity of generated molecules or reproduce historical biases in pharmaceutical development. We encourage future work to address these concerns through development of responsible use protocols, integration of toxicity and safety prediction tools, and careful curation of training datasets to ensure equitable representation across chemical domains. The machine learning community should collaborate with domain experts in chemistry, biology, and ethics to ensure that advances in molecular generation technologies maximize societal benefit while minimizing potential harms.

## K  Limitations

Despite the many documented advantages of DMol in terms of efficiency and molecular quality, several limitations remain. First, our approach still struggles with generating certain complex motif structures and stereochemistries, which are crucial for therapeutic applications. Second, while we achieve high validity scores, we do not explicitly enforce chemical constraints during generation, occasionally producing molecules that are formally valid but synthetically impossible to bring into existence. Third, our evaluation focuses primarily on small drug-like molecules; the performance on larger biomolecules, polymers, or metal-organic frameworks remains unexplored. Finally, the model's adaptability to conditional generation tasks (e.g., optimizing for specific target properties) requires additional architectural modifications that may affect the established efficiency gains. Future work should address these limitations to further bridge the gap between computational generation and practical chemical synthesis. Most importantly, this and all other related works should put more effort in identifying more comprehensive evaluation metrics for the generated molecules, since an overwhelming number of created samples cannot be synthesized or do not dock on any known protein.

## L  Future Work

Future work on DMol will focus on extending the model to conditional molecule generation, where specific chemical properties can be targeted through controlled diffusion processes. We also plan to address the remaining stereochemistry challenges by incorporating 3D structural information into the diffusion process. Additionally, we aim to develop more sophisticated motif compression strategies that can handle a wider variety of molecular scaffolds while preserving chemical feasibility. Exploring the application of DMol to larger biomolecular structures and integrating it with downstream property prediction models presents promising research directions. Finally, we intend to investigate dynamic scheduling of diffusion steps based on molecular complexity to further optimize computational efficiency.

