# OpenReview forum: "DMol: A Highly Efficient and Chemical Motif-Preserving Molecule Generation Platform"
_NeurIPS.cc/2025/Conference — NeurIPS 2025 poster_

### Official Review · Reviewer_FMwP · 2025-06-29

**Clarity:** 3
**Significance:** 3
**Originality:** 3
**Rating:** 4
**Confidence:** 1

**Summary:**

This paper targets the problem of small drug molecule generation. The paper proposes DMol (Diffusion Models for Molecular Motifs), which reduce the computational cost of diffusion models by relating the diffusion time-steps with the number of nodes and edges. DMol applies node compression that allows one to convert chemically important node motifs into supernodes. Experimental results show the effectiveness of the proposed method on small drug molecule generation tasks.

**Questions:**

* How is DMol compared to FragFM[1]?

[1]Lee et al.,  FragFM: Hierarchical Framework for Efficient Molecule Generation via Fragment-Level Discrete Flow Matching

**Ethical Concerns:**

["NO or VERY MINOR ethics concerns only"]

**Limitations:**

yes

**Quality:**

3

**Strengths And Weaknesses:**

pros:
* DMol introduces a hybrid noise schedule that balances deterministic perturbations with random noise. This approach improves motif preservation and ensures a flexible control of the diffusion process, resulting in better molecule quality.
* DMol introduces a novel "ring compression" approach, enabling the preservation and integration of chemically significant substructures like carbon rings. This method outperforms existing methods like DiGress and DeFoG in maintaining chemical motifs, which are crucial for practical drug design.
* Experiments show the emprical gain of DMol on small drug molecule generation tasks. DMol achieves a 3% improvement in SMILES validity over DiGress and a 4% improvement in motif-conservation scores compared to DeFoG. It also highlights the limitations of SMILES validity as a sole evaluation metric, focusing on more meaningful measures like chemical motif preservation.

cons:
* None

---

> ### Author Rebuttal · Authors · 2025-07-30
>
> We thank the reviewer for the constructive suggestion.
>
> FragFM is indeed a very recent paper that has not been published yet (to the best of our knowledge). We were unable to locate publicly available code for FragFM, which limits our ability to conduct a direct empirical comparison at this time. We will definitely add a citation to FragFM in our revised manuscript and acknowledge it, and, if time permits, try to implement their software. Also, upon reviewing the FragFM paper content, we note that while both DMol and FragFM address molecular generation, they employ fundamentally different methodologies: FragFM utilizes fragment-level discrete flow matching with a hierarchical coarse-to-fine autoencoder and stochastic fragment bag strategies, whereas DMol employs diffusion models with ring compression, codependent node-edge perturbations with deterministic number of changes per step which give rise to valuable loss penalties, and specialized noise schedules for motif preservation. We observe that FragFM's use of discrete flow matching is related to DeFoG, which we already included as a baseline in our comparisons.
>
> Thank you again for the extensive feedback on our work and for raising such interesting questions.

---

> ### Comment · Area_Chair_TCSc · 2025-08-09
>
> Reviewer FMwP, submitting a “Mandatory Acknowledgement” without leaving any comments for the authors in the discussion is not allowed. Could you provide feedback on the authors’ responses to your review comments? AC

---

### Official Review · Reviewer_iPx9 · 2025-07-03

**Clarity:** 3
**Significance:** 2
**Originality:** 2
**Rating:** 5
**Confidence:** 2

**Summary:**

The manuscripts presents two modifications of the DiGress method, a graph generative model:
1.) the replacement of common substructures, mainly carbon rings, by supernodes
2.) instead of swapping each atom in a graph every diffusion step by some probability, the method presented her suggest to first sample a subgraph in each diffusion step and then only swap atoms and edges in this subgraph

Both modifications aim to speed up the graph diffusion and improve its validity rate.

**Questions:**

- Can you comment on the possibility and probability that some edges are never changed in the forward process in case they are never part of the a induced subgraph?
- It would improve the manuscript if also a conditional generation benchmark would be included, although that is probably a lot of work

**Ethical Concerns:**

["NO or VERY MINOR ethics concerns only"]

**Final Justification:**

The authors could convince me that the supernodes do not introduce a strong bias in the generative model. I do not share other concerns about running times are not important raised by other referees - in my opinion the running time improvements might be important in some future applications we might not have in mind yet. Overall, this seems to be a valuable improvement over the Digress method.

**Limitations:**

yes

**Paper Formatting Concerns:**

Nothing

**Quality:**

3

**Strengths And Weaknesses:**

Weakness:
- the replacement of motifs by supernodes is an old trick
- these supernodes have the side effect that graph diffusion might produce with different sizes
- the choice of motifs might have a huge impact on the generation process, a few examples:
  - if a certain ring structure is part of a motif (say, a chlorinated carbon ring) and a certain ring structure is not (say bromated ring), it is not possible during diffusion to swap from one ring to the other. This might also mess up the probabilities with which certain substructures are generated
- using different set of motifs for different datasets/benchmarks is also not very convincing
- there are many aspects in DMol that are not entirely clear to me. For example, when only egdges of induced subgraphs are changed, there might be some or a lot of edges that are never changed during the whole diffusion process (i.e. when two nodes never occur in the same sampled subset).
- the benchmarks only cover unguided diffusion and evaluate the novelty, uniqueness and molecule-likeness of the outputs. It is not shown that DMol also outperforms DiGress on property optimization benchmarks. However, also the original DiGress paper did only perform a single benchmark on conditional generation

Strength:
- One issue with DiGress is that atom class marginal distributions are very skewed with most atoms are carbon and, thus, most transitions are carbon to carbon and have no effect. Therefore, DiGress is doing very few modifications per diffusion step and needs a lot of steps to add enough noise to a graph. The method here claims to need less steps due to their different sampling strategy. As I understand it, the nodes in the sampled subgraph all have to change their state, so there are no transitions from, say, Carbon to Carbon
- the number of steps in DMol is two times the number of nodes in the graph which is usually a magnitude less steps than DiGress
- sample time seems to be two magnitudes faster than DiGress
- compressing common motifs to supernodes might preserve these motifs better in the generated output. However, it feels like a poor workaround, as a good diffusion model should be able to preserve such motifs without such tricks.

---

> ### Author Rebuttal · Authors · 2025-07-30
>
> We thank the reviewer for constructive suggestions. The letter.number W.X and Q.X in our response refers to the bullet listed as weakness and question, followed by its number.
>
> W1: We respectfully disagree with the “old trick” characterization. While concepts similar to supernodes have been employed in methods like JT-VAE, our approach addresses fundamental limitations of VAEs. JT-VAE requires minimum spanning tree construction during encoding and faces  significant challenges in decoding due to the complexity and ambiguities that need to be resolved when reattaching molecular substructures with multiple bond types to scaffolds. JT-VAE also uses several hundred supernodes, which is unacceptable for diffusion models. Furthermore, the authors of DiGress clearly stated that their attempts at supernode/motif masking failed (see their GitHub issue #15, and they remark that “It (meaning masking) would probably work better if we preserved the motif during diffusion in training,” but they did not come up with a satisfactory resolution). In contrast to JT-VAE, our method only uses 15 carbon ring motifs that can always be reintegrated into the scaffold in a chemically valid manner (our selection was based on ring frequencies and was done in consultation with chemists). Note that these motifs change for each dataset as different sets of drugs do have different motifs (e.g., high frequency substructures).
>
> W2: For DMol, there is no concern pertaining to different molecular sizes since this issue is addressed through its design which preserves the *joint* distribution of individual atoms and motifs. Our approach does not arbitrarily replace motifs with single atoms, but motifs with other motifs or  single atoms, because we explicitly model and preserve the distribution of motifs in the training samples. The diffusion process reconstructs both individual atoms and motifs according to their empirical distributions, ensuring that molecules maintain appropriate structural complexity. Additionally, our model incorporates a unique feature where the number of diffusion steps depends on the size of the molecule ($T = 2n$), and our objective function accounts for this relationship through penalty terms that ensure consistent node and edge count modifications. This design prevents problematic size shrinkage.
>
> W3: It appears that there is a misunderstanding about our approach. While our predefined motifs (such as chlorinated carbon rings) cannot be  directly transformed into nonmotif rings (such as brominated rings) during diffusion, this limitation does not impair our model's generative capabilities. Rings that are not used as motifs, of which there are many,  can still be generated through atom-by-atom and edge-by-edge modifications during standard atom diffusion, which is how ours and other diffusion models manage to generate new ring structures in the first place. For example, a brominated ring can be generated by modifying an individual chlorine atom to a bromine atom through single-node state transitions. Our hybrid approach preserves the probability distributions of both motif and nonmotif structures. The key idea is that motif compression enhances generation of frequent rings while retaining full flexibility for generating arbitrary molecular patterns through an atom-by-atom level diffusion process.
>
> W4. We respectfully disagree with the claim. By definition, a motif is a substructure that appears with significantly higher frequency than would be expected by random chance in a given dataset. Different molecular datasets naturally exhibit different motif structures/frequencies because they represent distinct chemical compounds with varying biological activities and synthetic accessibility constraints. Hence, MOSES, Guacamole etc all have different motifs (highest frequency substructures). As stated in line 302 of our paper, we systematically select the most frequently occurring carbon rings that can always form single bonds with scaffolds through available carbons. This dataset-specific selection is not only appropriate but necessary to capture the underlying chemical diversity of datasets.
>
> W5 & Q1: There may be a misunderstanding how our forward diffusion process operates. In Appendix B (lines 669-679), we provide a comprehensive comparison of the number of edges modified at each time step of DMol and DiGress. In our approach, since the selection of a *deterministic number* of nodes at each time step is random, every edge in the graph has a nonzero probability of being selected and modified during the diffusion process. The probability that any edge remains unchanged throughout the entire diffusion process is the product of these probabilities across all time steps and our number of steps ensures that each edge has a nonzero probability of being selected. It is nevertheless important to understand that no diffusion model will or has to modify every node and edge.
>
> Table 1: The result of conditional generation on MOSES dataset
> | Method | Validity | Uniqueness | Novelty | Filters | FCD | SNN | SCAF | ChEMBL | QED |
> |--------|----------|------------|---------|---------|-----|-----|------|--------|-----|
> | First collector | 82.5 | 100 | 100 | 93.1 | 2.05 | 0.47 | 13.9 | 4.212 | 0.7842 |
> | Second collector | 83.1 | 100 | 100 | 93.0 | 1.93 | 0.48 | 13.6 | 4.1873 | 0.7885 |
> | Third collector | 82.3 | 100 | 100 | 93.2 | 1.94 | 0.45 | 13.7 | 4.1796 | 0.7921 |
> | Fourth collector | 81.8 | 100 | 100 | 93.6 | 1.99 | 0.43 | 13.0 | 4.2017 | 0.7804 |
> | Fifth collector | 80.7 | 100 | 100 | 93.5 | 1.92 | 0.48 | 13.7 | 4.1121 | 0.7882 |
> | Aromatic monocyclic | 77.6 | 100 | 100 | 89.2 | 2.23 | 0.43 | 12.4 | 3.8313 | 0.7348 |
> | Aromatic monocyclic heterocycle | 76.1 | 100 | 100 | 88.7 | 2.37 | 0.43 | 12.8 | 3.754 | 0.7257 |
> | Fused bicyclic | 75.9 | 100 | 100 | 87.3 | 2.31 | 0.40 | 12.2 | 3.6008 | 0.7233 |
> |Aromatic heterocyclic | 75.8 | 100 | 100 | 87.4 | 2.19 | 0.45 | 12.7 | 3.9126 | 0.7109 |
>
> W6& Q2: Regarding conditional generation, we actually did multiple experiments on this topic already. Rather than resort to complicated and time-consuming conditional generation, we opted instead to classify the molecules based on their scaffold (using four classes, aromatic monocyclic, aromatic monocyclic heterocycle, fused bicyclic, aromatic heterocyclic, following RDKit recommendations) or based on their graph embeddings (obtained by using GraphSage GNNs, followed by Kmeans++ clustering). In the latter case, we selected five clusters with the largest number of samples and best cluster separations. The results(Table 1) demonstrate that DMol perform significantly better on the GNN-governed clusters (validity: 80.7-83.1%, ChEMBL: 4.11-4.21) compared to scaffold based clusters (validity: 75.8-77.6%, ChEMBL: 3.60-3.91). This is a consequence of the fact that  RDKit scaffold classes exhibit significant overlaps, while the GNN-based classes are well-separated. The GNN-based results also show slight degradations in performance compared to the full datasets which may be attributed to a significant reduction in the number of training samples.
>
> Thank you again for the extensive feedback on our work and for raising such interesting questions.

---

> > ### Author Response · Authors · 2025-08-05
> >
> > We once again thank the reviewer for insightful comments and questions. If there are some points in our rebuttal that need further elaboration, please do let us know.

---

> > > ### Comment · Reviewer_iPx9 · 2025-08-06
> > >
> > > Regarding my example with the halogenated ring: yes, in principal diffusion is able to transform one ring into another, but only by using other atoms to form the ring. It is, for example, not possible to diffuse one molecule with a chlorine ring into the very same molecule but with a bromine ring - both representations have different number of (super)nodes and the diffusion method presented here cannot change the number of nodes. Also, my critique is not so much that certain substructures are "impossible" to produce, but instead that the probabilities to generate certain substructures might be messed up by introducing these supernodes. Using the example above: the probability that a chlorine ring is formed by pure chance (or at late timestep) is relatively high (e.g., corresponds to the frequency of chlorine rings in data) while the same is not true for a bromine ring. While this is more or less the whole idea of the introduction of supernodes (making frequent structures more easy to generate) it might also introduce a strong bias for some substructures. In particular, that might effect substructures which are frequent but not frequent enough for being supernodes (it's a hard threshold in the end). I wonder if such an effect could be visualized by counting substructures in the generated molecules and comparing substructures that are frequent in the training data between motif and non-motif. I suspect the former being much more frequent than the later.

---

> > > > ### Author Response · Authors · 2025-08-07
> > > >
> > > > We thank the reviewer for the insightful feedback and expressing the concern regarding potential distortion of subgraph/motif probabilities due to the use of supernodes. With regards to your question regarding the impossibility to transform a certain structure into another (e.g., a chlorinated into a bromated ring): We need to revisit the definition of a motif. In the case of chlorinated and bromated rings, if both are frequent in the dataset, then the benzene rings that are common to both structures are motifs as well, since their frequency is higher than that of each of  the “decorated rings”. Hence, we would have the benzene substructure to work with as well, and this motif can be augmented by other atoms through the diffusion process via additions of new bonds and removal of others (we are running the diffusion process on both edges and nodes). Hence, the reviewer is correct that if only the two decorated aromatic rings were motifs, conversion would be impossible, but the point is that the shared ring structure would in this case also have to be a motif (as our definition of motifs is based on their frequency of occurrence). Consequently, redecoration would enable the aforementioned transform.
> > > >
> > > > To directly address this concern, we conducted an analysis comparing the probabilities of motifs in the training and generated sets. Specifically, we examined the top 30 most frequent motifs: the top 15 of these are used as supernodes in our model, while the remaining 15 serve as a comparison group. For each motif, we computed the probability that a molecule in a given set contains that motif. Note again that the reported values represent the probability that a molecule contains a given motif, and hence the values in the individual columns do not represent distributions (the probabilities across different motifs do not sum up to one), since a single molecule can contain multiple motifs or none.
> > > >
> > > > Due to space limitations, we present results for the MOSES dataset but will include additional results for Guacamol in the final paper (the results are very similar). The table below shows the per-motif probabilities in the training and generated sets.
> > > >
> > > > | ID | Motif       | Prob on Training Set | Prob on Generated Set |
> > > > |-----|--------------|----------------|----------------|
> > > > | 0   | c1ccccc1     | 0.775340       | 0.786094       |
> > > > | 1   | c1ccncc1     | 0.182872       | 0.163003       |
> > > > | 2   | c1cnnc1      | 0.101933       | 0.091161       |
> > > > | 3   | C1CCNCC1     | 0.085190       | 0.078320       |
> > > > | 4   | C1CCNC1      | 0.079766       | 0.066173       |
> > > > | 5   | c1cscc1      | 0.076320       | 0.074040       |
> > > > | 6   | c1ccsn1      | 0.071510       | 0.068602       |
> > > > | 7   | C1COCCN1     | 0.062541       | 0.065132       |
> > > > | 8   | C1CNCCN1     | 0.061973       | 0.068140       |
> > > > | 9   | c1ccoc1      | 0.057304       | 0.042920       |
> > > > | 10  | c1cncnc1     | 0.055889       | 0.051249       |
> > > > | 11  | c1cncn1      | 0.046374       | 0.041647       |
> > > > | 12  | c1ncon1      | 0.045917       | 0.045812       |
> > > > | 13  | c1ncnn1      | 0.045303       | 0.041532       |
> > > > | 14  | C1CCCCC1     | 0.035119       | 0.030079       |
> > > > | 15  | C1CC1        | 0.029827       | 0.024179       |
> > > > | 16  | c1cnoc1      | 0.028895       | 0.023832       |
> > > > | 17  | C1CCCC1      | 0.024560       | 0.028112       |
> > > > | 18  | C1CCOC1      | 0.023108       | 0.024063       |
> > > > | 19  | c1ccnc1      | 0.021390       | 0.019088       |
> > > > | 20  | c1cCNCC1     | 0.020182       | 0.016890       |
> > > > | 21  | c1ccnnc1     | 0.019092       | 0.012263       |
> > > > | 22  | c1nnnn1      | 0.017667       | 0.016081       |
> > > > | 23  | c1nncs1      | 0.016355       | 0.014345       |
> > > > | 24  | c1cnccn1     | 0.015449       | 0.012726       |
> > > > | 25  | c1cnnn1      | 0.015114       | 0.010412       |
> > > > | 26  | c1cocn1      | 0.014555       | 0.008792       |
> > > > | 27  | c1nnco1      | 0.013514       | 0.009024       |
> > > > | 28  | c1cCCCC1     | 0.012324       | 0.009833       |
> > > > | 29  | c1cNCC1      | 0.011438       | 0.010180       |
> > > >
> > > > Average probability:
> > > >
> > > > |                     | Prob on Training Set | Prob on Generated Set |
> > > > |---------------------|--------------------------|----------------------------|
> > > > | Top 15 mean         | 0.1189                   | 0.1143                     |
> > > > | Last 15 mean        | 0.0189                   | 0.0160                     |
> > > >
> > > > As may be seen, the distributions are closely aligned, and we observe no significant gap between the supernode and nonsupernode groups, indicating that our approach does not distort the overall motif distribution.
> > > >
> > > > Additionally, we examined the joint probabilities of the two most frequent motifs and other pairs of motifs. For example, the probabilities of seeing both benzene (c1ccccc1) and pyridine (c1ccncc1) co-occurring in the same molecule are
> > > >
> > > > Prob on Training Set : 0.11
> > > >
> > > > Prob on Generated Set:  0.1
> > > >
> > > > We will add a complete table into our revision to illustrate that our model maintains realistic co-occurrence patterns.

---

> > > > > ### Author Response · Authors · 2025-08-07
> > > > >
> > > > > Finally, we compared the marginal distributions of node/supernode categories (note that supernodes are treated as new node types, so all probabilities in this case sum up to one) between molecules from the MOSES training set and newly generated molecules after ring compression. The results are shown below:
> > > > >
> > > > > | Nodes / SuperNodes | Training Set | Generated Set |
> > > > > |--------------------|--------------|---------------|
> > > > > | C                  | 0.4897       | 0.4940        |
> > > > > | N                  | 0.1511       | 0.1506        |
> > > > > | S                  | 0.0170       | 0.0167        |
> > > > > | O                  | 0.1654       | 0.1644        |
> > > > > | F                  | 0.0246       | 0.0243        |
> > > > > | Cl                 | 0.0093       | 0.0091        |
> > > > > | Br                 | 0.0025       | 0.0026        |
> > > > > | c1ccccc1           | 0.0736       | 0.0750        |
> > > > > | c1ccncc1           | 0.0132       | 0.0130        |
> > > > > | c1cnnc1            | 0.0081       | 0.0077        |
> > > > > | C1CCNCC1           | 0.0068       | 0.0070        |
> > > > > | C1CCNC1            | 0.0049       | 0.0045        |
> > > > > | c1cscc1            | 0.0055       | 0.0052        |
> > > > > | c1ccsn1            | 0.0056       | 0.0050        |
> > > > > | C1COCCN1           | 0.0035       | 0.0036        |
> > > > > | C1CNCCN1           | 0.0036       | 0.0031        |
> > > > > | c1ccoc1            | 0.0041       | 0.0039        |
> > > > > | c1cncnc1           | 0.0044       | 0.0040        |
> > > > > | c1cncn1             | 0.0010       | 0.0008        |
> > > > > | c1ncon1            | 0.0027       | 0.0025        |
> > > > > | c1ncnn1             | 0.0024       | 0.0020        |
> > > > > | C1CCCCC1           | 0.0010       | 0.0008        |
> > > > >
> > > > >
> > > > > The results demonstrate that the marginal distributions between the training set molecules and newly generated molecules are close. This aligns with the fundamental goal  of diffusion models - namely, to sample from the original data distribution. Therefore, diffusion models like DMol and Digress essentially control the consistency between: (1) the distributions of atom types and bond types in molecules generated from noise, and (2) those in the training set molecules.
> > > > >
> > > > > We attribute this strong preservation of substructure statistics to two key design steps:
> > > > > Sampling based on marginal distributions: We use the marginal distribution of motifs in the training set as the prior for supernode sampling, and similarly, the marginal distribution of atom types for node sampling. This ensures that both motif-based and nonmotif subgraphs are generated with probabilities consistent with the training data.
> > > > > Diversity in sampling steps: We enforce that 𝑁(𝑡) nodes are changed at each time step, ensuring that substructures beyond the predefined motifs have sufficient opportunities to be generated through multi-node sampling.
> > > > >
> > > > > Once again, we appreciate the reviewer’s question and agree that maintaining substructure fidelity is crucial. Our results demonstrate that the proposed method effectively preserves both individual motif distributions and broader structural patterns. We hope this helps with alleviating any remaining concerns.

---

### Official Review · Reviewer_6j7c · 2025-07-05

**Clarity:** 3
**Significance:** 3
**Originality:** 3
**Rating:** 3
**Confidence:** 5

**Summary:**

This paper introduces a novel graph diffusion model for molecule generation with fast sampling speed and motif-preserving property. In detail, it modifies the sampling steps w.r.t. the number of atoms for accelerating sampling and retain the dependency between nodes and edges in the forward process. Moreover, it develops a ring compression technique to preserve chemical motifs along the diffusion process.

**Questions:**

On Line 245, DMol samples the nodes and edges from specific marginal distributions. Why not introduce the motivation of the choice of the prior distributions?
Since the paper mentions the JT-VAE on line 307, why doesn’t it compare with this model in the experiments?
One of your strengths is fast sampling. Why not plot a figure to show your sampling speed compared to other methods?
One of your strengths is motif-preserving. Why not show some experimental results in the main text to show this property?
Why not introduce your model’s name in the abstract?

**Ethical Concerns:**

["NO or VERY MINOR ethics concerns only"]

**Final Justification:**

I appreciate the authors' rebuttal, which has addressed most of my concerns. At this stage, I intend to maintain my original score and would be interested in seeing the feedback from the other reviewers.

**Limitations:**

Yes

**Quality:**

4

**Strengths And Weaknesses:**

Strengths
Providing deep analysis of sampling efficiency and proving that DMol would be more efficient than Digress when changing a node.
It proves the fast sampling properties of the proposed model empirically.
The paper is clearly written. It first introduces the framework of graph diffusion models, then develops on this base, which is clear for the readers.

Weaknesses
Technically impressive, but of limited practical value. This work brings little improvement in validity and ChEMBL-likeness and accelerates the sampling process by a large margin, compared to other methods. However, (i) the wet-lab experiment takes much more time than the in silico one, (ii) the sampling time of current methods is acceptable (e.g. DEFOG 5.81s). It means that the sampling time should not be the bottleneck in drug development.
 Poor formatting. The authors try to plug too much information into the main text, resulting in a tight page layout (the algorithms, the equations, etc). Besides, the font size in Figure 1 is too small to read.
Ambiguous notations. On Line 230, the paper uses index j to represent both state and class, which may lead to confusion. It’s better to assign the indexes properly throughout the paper.
Over-simplified metrics on experimental tables. It seems that there is enough space for the tables to write down some of the full names of the evaluation metrics. For instance, the V, U, N metrics are really indirect for the readers who only read the tables.

---

> ### Author Rebuttal · Authors · 2025-07-30
>
> We thank the reviewer for insightful comments and for taking the time to review our work.
>
> ### Weakness
> > Technically impressive, but of limited practical value. This work brings little improvement in validity and ChEMBL-likeness and accelerates the sampling process by a large margin, compared to other methods. However, (i) the wet-lab experiment takes much more time than the in silico one, (ii) the sampling time of current methods is acceptable (e.g. DEFOG 5.81s). It means that the sampling time should not be the bottleneck in drug development.
>
> We thank the reviewer for acknowledging the large improvements in validity, ChEMBL-likeness, and sampling speed of DMol. We agree that wet-lab experiments dominate the timeline during the later stages of drug development. However, sampling efficiency remains a critical factor that influences early-stage drug design pipelines, particularly in the de novo hit-to-lead phase where no prior reference compounds are available for a target protein. This is especially the case for high-throughput screening scenarios, for which the availability of a very large number of diverse and chemically valid candidates is essential.
>
> For example, among molecules generated by all diffusion models including DiGress and DMol, those with a ChEMBL-likeness (CLscore) above 5.5 account for fewer than 0.5% of total outputs. While DMol significantly outperforms other models in this regard, each sampling run typically yields only a few dozen molecules that meet minimal thresholds for chemical likeness (note that scores closer to 8/9 are desirable, yet still not sufficient guarantees that the molecules can actually be synthesized). For example, Contract Research Organizations (CROs) such as Evotec and Nuvisan offer high-quality drug compound libraries containing over one million molecules, which are purchased and screened by pharmaceutical and biotech companies and it is easy to see that to get to such numbers purely through AI generation poses significant time constraints. This highlights the scale of high-throughput requirements and the need to generate large volumes of candidate molecules in de novo pipelines.
>
> Hence, improvements in sampling speed directly enable more comprehensive chemical space exploration. Although methods like DeFoG already demonstrate efficient generation times (e.g., 5.81s/64 molecule), one has to note, as pointed above, that most of the molecules will not be chemically valid to the point that they can be synthesized. DMol offers further acceleration as well as a significantly better chance that the molecules will be valid due to their motif conservation. We believe this performance gain is meaningful in practical settings, especially when millions of candidates must be explored.
>
> In conclusion, the general goal is not to merely generate as many molecules as possible, but to generate high-quality molecules, which is an inherently difficult task. This is not a critique of DeFoG or any other method specifically, but rather a shared limitation among current generative approaches in the field.
>
> > Poor formatting. The authors try to plug too much information into the main text, resulting in a tight page layout (the algorithms, the equations, etc). Besides, the font size in Figure 1 is too small to read. Ambiguous notations. On Line 230, the paper uses index j to represent both state and class, which may lead to confusion. It’s better to assign the indexes properly throughout the paper.
>
> We thank the reviewer for this helpful feedback and apologize for the lack of readability. We acknowledge that we included a large amount of technical content in the main text. Given the complexity of drug discovery—requiring both rigorous mathematical formulation and deep domain knowledge—we hoped that including all core ideas would demonstrate the multifaceted ideas we tried to integrate to make the model more successful. We will nevertheless follow the advice and revise the formatting to improve clarity. Regarding line 230, the state j and class j are both referring to node type j. We apologize for this ambiguous use of the same symbol.
>
> > Over-simplified metrics on experimental tables. It seems that there is enough space for the tables to write down some of the full names of the evaluation metrics. For instance, the V, U, N metrics are really indirect for the readers who only read the tables.
>
> We will revise all the tables to include the full names of evaluation metrics, improving readability for readers unfamiliar with our shorthand notation.
>
> ### Questions
> > On Line 245, DMol samples the nodes and edges from specific marginal distributions. Why not introduce the motivation of the choice of the prior distributions?
>
> In standard diffusion models, Gaussian (for continuous) or uniform (for discrete) priors are commonly used. However, in molecular generation tasks, such priors can be suboptimal due to the highly skewed distribution of atom and bond types in real-world molecules (e.g., carbon atoms are predominant in molecular graphs). Using a uniform prior significantly increases the learning burden on the model, as it must transform a flat distribution into a highly structured one. By contrast, using the empirical marginal distributions as priors gives the model a better inductive starting point, improving convergence and sample quality. This approach is supported by the recommendation in the landmark paper DiGress (see Section 4 and Table 6), where marginal priors resulted in better performance than uniform ones. Note that our priors are marginals pertaining to both atoms and the selected motifs.
>
> > Since the paper mentions the JT-VAE on line 307, why doesn’t it compare with this model in the experiments?
>
> JT-VAE is a landmark model that has been widely used for benchmarking in earlier molecular generation work. However, it is based on VAEs, which operates in a latent space and requires highly complex decoding procedures to ensure chemical validity, such as enforcing valency constraints through hard-coded rules. DiGress also highlights these “hard-coded rules” in Table 2, which we believe makes direct comparison with diffusion-based models unfair. In contrast, our method—and the baselines we compare against—are based on more recent paradigms such as diffusion models and flow matching. To ensure a fair and up-to-date comparison, we followed the common practice in the field by benchmarking against these more recent and directly comparable diffusion approaches.
>
> > One of your strengths is fast sampling. Why not plot a figure to show your sampling speed compared to other methods?
>
> We appreciate the reviewer’s suggestion. In our submitted paper, we indeed presented the sampling times (rather than speeds) of each method in Table 4. We believe that this table provides a direct and quantitative comparison and clearly demonstrates the efficiency of DMol.
>
> > One of your strengths is motif-preserving. Why not show some experimental results in the main text to show this property?
>
> We thank the reviewer for highlighting DMol’s motif-preserving capabilities. Since molecular motifs (also known as shingles according to RDKit) play a crucial role in determining the chemical properties of a molecule, metrics such as ChEMBL-likeness and QED are specifically designed to reflect how well a molecule captures and preserves important biomedical shingles/motif substructures. For a more detailed explanation of the ChEMBL-likeness and QED metric, please refer to Appendix F of our paper.
>
> > Why not introduce your model’s name in the abstract?
>
> We thank the reviewer’s suggestion - we will add the model’s name in the abstract for better recognition of the method.
>
> Thank you again for the extensive feedback on our work and for raising such interesting questions.

---

> > ### Author Response · Authors · 2025-08-05
> >
> > We once again thank the reviewer for insightful comments and questions. If there are some points in our rebuttal that need further elaboration, please do let us know.

---

> ### Comment · Area_Chair_TCSc · 2025-08-09
>
> Reviewer 6j7c, not responding to the authors / submitting a “Mandatory Acknowledgement” without posting a single comment to the authors in the discussion is not permitted. Could you provide feedback on the authors’ responses to your review comments? AC

---

### Official Review · Reviewer_pDy8 · 2025-07-21

**Clarity:** 4
**Significance:** 2
**Originality:** 3
**Rating:** 5
**Confidence:** 4

**Summary:**

This work proposes a novel graph diffusion model (DMol) for small molecule generation task that enhances semantic controllability and structural motif preservation. Motivated by the limitations of prior discrete diffusion models such as DiGress [1], the work introduce three key innovations:

1. Forward Noise Injection: At each diffusion step, DMol perturbs only a deterministic number of nodes and their associated edges within an induced subgraph, using learned transition matrices.

2. Denoising and Sampling: The model incorporates auxiliary functions that encode the expected level of perturbation at each time-step, guiding the denoising network toward more accurate reconstructions. A modified and permutation-invariant loss penalizes deviations from the scheduled number of node and edge changes.

3. Ring Compression: Typical carbon ring motifs are encoded as super-nodes, allowing for explicit preservation of chemically meaningful structural motifs and efficient decoding into valid molecular graphs.

Experiments conduced on multiple benchmark datasets demonstrate that DMol achieves competitive or superior performance compared to multiple state-of-the-art baselines. Besides standard set of V.U.N. (Validity, Uniqueness, Novelty) metrics, this work also measured chemically informed metrics, e.g., ChEMBL-Likeness Score [2], Quantitative Estimation of Drug-likeness (QED) [3], and a newly proposed Shingle Distance (SD).

[1] Vignac, Clement, et al. "Digress: Discrete denoising diffusion for graph generation." arXiv preprint arXiv:2209.14734 (2022).

[2] Gaulton, Anna, et al. "The ChEMBL database in 2017." Nucleic acids research 45.D1 (2017): D945-D954.

[3] Bickerton, G. Richard, et al. "Quantifying the chemical beauty of drugs." Nature chemistry 4.2 (2012): 90-98.

**Questions:**

Please refer the review in weakness.

**Ethical Concerns:**

["NO or VERY MINOR ethics concerns only"]

**Final Justification:**

My final justification is still 5: Accept.

**Limitations:**

yes.

**Paper Formatting Concerns:**

No formatting issues.

**Quality:**

3

**Strengths And Weaknesses:**

Strengths:
1. Methodologically novel and chemically informed diffusion framework that incorporates principled constraints on chemical motifs, enabling structure-aware and efficient molecule generation.
2. Chemically-justified ring compression method for important motif / scaffold preservation (consistently overlooked by prior similar generation task, to the best of my knowledge).
3. Comprehensive benchmarking with prior models.
4. The authors made strong efforts to incorporate more chemically meaningful metrics that go beyond standard validity measures.

Weaknesses:
1. The only major concern to this paper roots from its Significance. While the method improves on small molecule (<40 atoms) generation metrics, it lacks validation on actual Computer-aided Drug Design-related downstream tasks, so the gap between generation quality and real-world utility remains unaddressed.
2. No direct ablation study is observed for the forward process cosine schedules. It might remains unclear whether this scheduling imposes constraints across molecules with varying scaffolds or topologies.

---

> ### Author Rebuttal · Authors · 2025-07-30
>
> We thank the reviewer for constructive suggestions. The letter.number W.X in our response refers to the bullet listed as weakness, followed by its number.
>
> W1. We completely agree that the validity of generated molecules should ultimately be judged based on their utility in downstream pharmaceutical tasks. Unfortunately, the generative AI community as a whole currently does not have validation metrics that can be used to guarantee downstream performance. As a small step in this direction, we would like to draw your attention to the downstream validation results presented in our Supplement. For proprietary reasons, we cannot comment about our target selection rationale, but we indeed evaluated the AI-generated drugs as inhibitors of  two proteins, 1PXH (Protein-tyrosine phosphatase) and 4MQS (human M2 muscarinic acetylcholine receptor). We ran DMol and identified molecules with fairly large  CheMBL scores (>5) and certain desirable similarity with known drugs targeting the same proteins. We then ran docking simulations with AutoDock Vina (a simulation software that is not as accurate as an experiment, yet still one of the most sophisticated docking methods available). As shown in Figures 12 and 13 of the Supplement, our generated drugs illustrate both the strengths and limitations of AI approaches. For the 1PXH target (Figure 12), our selected generated molecules showed higher (less favorable) free energies (-6.4 and -6.0 kcal/mol) compared to the reference drug (-8.7 kcal/mol), which may be, among other reasons, attributed to the small molecular size of the drug candidate compared to the size of the reference inhibitor. We therefore focused on generated molecules that have similar sizes, motifs and compositions as the reference drugs and for the 4MQS target (Figure 13), we ended up with a molecule that  significantly outperformed the best reference, achieving a free energy of -9.1 kcal/mol compared to the reference’s -6.4 kcal/mol. As a result, we made sure that for each target protein we sub-selected AI-generated drugs with appropriate molecular mass and motif structures. We chose not to emphasize this test in the main text due to space limitations and in order to maintain the coherent structure of our exposition. However, we can easily comment on more sophisticated downstream validation approaches, including similarity metrics for the molecules based on their masses, motives, 3D shapes as well as their “distances” of embeddings (generated by graph neural network) from known references. We also had guidance from molecular dynamics simulations, but did not include the conclusions to once again maintain a coherent exposition. Bridging the gap between generation quality metrics and real-world utility is and always was our biggest priority.
>
> Table 1: The result of ablation study for the forward process schedules on MOSES dataset
> | Method | Validity | Uniqueness | Novelty | Filters | FCD | SNN | SCAF | ChEMBL | QED |
> |--------|----------|------------|---------|---------|-----|-----|------|--------|-----|
> | cos (c=0.004) | 86.9 | 100 | 100 | 97.4 | 1.13 | 0.57 | 14.2 | 4.5022 | 0.8022 |
> | cos (c=0.006) | 87.5 | 100 | 100 | 97.7 | 1.12 | 0.57 | 14.6 | 4.5020 | 0.8037 |
> | cos (c=0.008) | 87.8 | 100 | 100 | 97.8 | 1.12 | 0.58 | 14.8 | 4.5033 | 0.8055 |
> | cos (c=0.01) | 87.3 | 100 | 100 | 97.0 | 1.19 | 0.55 | 14.3 | 4.5018 | 0.8042 |
> | Linear | 85.2 | 99.8 | 99.9 | 96.1 | 1.19 | 0.56 | 14.2 | 4.5011 | 0.8016 |
>
> Table 2: The result of scaffold-constrained generation on MOSES dataset
> | Method | Validity | Uniqueness | Novelty | Filters | FCD | SNN | SCAF | ChEMBL | QED |
> |--------|----------|------------|---------|---------|-----|-----|------|--------|-----|
> | First collector | 82.5 | 100 | 100 | 93.1 | 2.05 | 0.47 | 13.9 | 4.212 | 0.7842 |
> | Second collector | 83.1 | 100 | 100 | 93.0 | 1.93 | 0.48 | 13.6 | 4.1873 | 0.7885 |
> | Third collector | 82.3 | 100 | 100 | 93.2 | 1.94 | 0.45 | 13.7 | 4.1796 | 0.7921 |
> | Fourth collector | 81.8 | 100 | 100 | 93.6 | 1.99 | 0.43 | 13.0 | 4.2017 | 0.7804 |
> | Fifth collector | 80.7 | 100 | 100 | 93.5 | 1.92 | 0.48 | 13.7 | 4.1121 | 0.7882 |
> | Aromatic monocyclic | 77.6 | 100 | 100 | 89.2 | 2.23 | 0.43 | 12.4 | 3.8313 | 0.7348 |
> | Aromatic monocyclic heterocycle | 76.1 | 100 | 100 | 88.7 | 2.37 | 0.43 | 12.8 | 3.754 | 0.7257 |
> | Fused bicyclic | 75.9 | 100 | 100 | 87.3 | 2.31 | 0.40 | 12.2 | 3.6008 | 0.7233 |
> |Aromatic heterocyclic | 75.8 | 100 | 100 | 87.4 | 2.19 | 0.45 | 12.7 | 3.9126 | 0.7109 |
>
> W2. To address the question regarding the cosine schedule, we reran DMol on the MOSES dataset using the cosine schedule with different values of  the parameter $c$ (mentioned in line 217 of our paper) as well as a linear schedule ($\alpha = t/T$). The experimental results are shown in Table 1. As can be observed from the experiments, varying the value of $c$ has small impact on the performance metrics, with the optimal performance achieved for $c=0.008$ (which is the result presented in our paper). This finding aligns with Digress's selection of the hyperparameter $c$, as its code also adopts $c=0.008$ as the default setting. When comparing cosine and linear schedules, the cosine schedule is clearly superior to linear schedules. This advantage primarily stems from the fact that the cosine schedule adds less noise per timestep during both the initial and final phases of the diffusion process compared to intermediate stages. This distribution better aligns with the learning characteristics of diffusion models. Specifically, maintaining relatively small noise levels during the early diffusion phase/the last diffusion phase facilitates the model's learning of fine-grained data features, and ensure smoother convergence. In contrast, the uniform noise distribution in linear schedules may prove either overly aggressive or overly conservative at certain stages, leading to compromised training efficiency and ultimate  performance.
>
> Regarding scaffold-constrained generation, we actually did multiple experiments on this topic already. Rather than resort to complicated and time-consuming conditional generation, we opted instead to classify the molecules based on their scaffold (using four classes, aromatic monocyclic, aromatic monocyclic heterocycle, fused bicyclic, aromatic heterocyclic, following RDKit recommendations) or based on their graph embeddings (obtained by using GraphSage GNNs, followed by Kmeans++ clustering). In the latter case, we selected five clusters with the largest number of samples and best cluster separations. The results demonstrate that DMol perform significantly better on the GNN-governed clusters (validity: 80.7-83.1%, ChEMBL: 4.11-4.21) compared to scaffold based clusters (validity: 75.8-77.6%, ChEMBL: 3.60-3.91). This is a consequence of the fact that  RDKit scaffold classes exhibit significant overlaps, while the GNN-based classes are well-separated. The GNN-based results also show slight degradations in performance compared to the full datasets which may be attributed to a significant reduction in the number of training samples. Furthermore, the excellent performance on individual GNN-classes shows that the cosine schedule does not impose special constraints for different scaffolds.
>
> Thank you again for the extensive feedback on our work and for raising such interesting questions.

---

> > ### Author Response · Authors · 2025-08-05
> >
> > We once again thank the reviewer for insightful comments and questions. If there are some points in our rebuttal that need further elaboration, please do let us know.

---

> > ### Comment · Reviewer_pDy8 · 2025-08-06
> >
> > First of all, my apologies for the delayed comment, and thank you very much for your efforts in addressing the weaknesses raised earlier.
> >
> >
> > Regarding W1: I fully agree that achieving the ultimate goal of computer-aided drug design across downstream tasks is far beyond the scope of a single paper. I truly appreciate your detailed response and the additional experiments demonstrating that the AI-generated molecules can act as inhibitors for the two target proteins (1PXH and 4MQS). The results appear solid and convincing, and I have no further questions on this point.
> >
> >
> > Regarding W2: The experimental design is clear, with comparison between cosine and linear schedules, as well as constraints to different scaffolds (no obvious constraints). As with W1, the setup makes sense to me, and I have no additional concerns.

---

> > > ### Author Response · Authors · 2025-08-07
> > > **Thank you for your response!**
> > >
> > > We very much appreciate you taking the time to carefully read the rebuttal.

---

### Note · Authors · 2025-08-12

We appreciate the constructive feedback provided by the AE and all reviewers.

Regarding Reviewer pDy8's comments on downstream validation and forward process scheduling, we addressed these by pointing to supplementary results, pertaining to AutoDock Vina docking free energies for 1PXH and 4MQS protein targets; **our generated molecules achieved superior binding affinity** (-9.1 kcal/mol vs -6.4 kcal/mol for the reference drug targeting 4MQS). We also provided a comprehensive ablation study for optimizing the cosine schedule and included scaffold-constrained generation results showing that constraints across molecular topologies still yield excellent validity/CLscore/QED.

Regarding Reviewer 6j7c's questions about the practical value of the work and motif preservation, beyond our 10-fold reduction in the number of diffusion steps, we showed significant increases in CLscore and QED compared to state-of-the-art diffusion methods. We also pointed out **the relevance of generation speed as it both allows for including more biochemical constraints and increasing the yield of high-quality molecules**. Our consistently higher CLscore and QED directly illustrate motif preservation, as these metrics measure biomedical motif/shingle substructure preservation. In addition (please see comment to Reviewer iPx9), we included results showing **supernode motif distributions remain highly preserved in the generated molecules**. The performance gains and execution time savings stem from new graph perturbation methods that jointly operate on a deterministic number of nodes and edges, careful selection of frequent subgraphs diffused jointly with atoms, and an adapted loss function. We had hoped to discuss any remaining concerns but did not have the opportunity.

Regarding Reviewer iPx9's concerns about motif compression and probability distortions. We demonstrated that **our approach maintains highly similar probability distributions of the original and generated data via motif analysis**. Comparing the top 30 most frequent motifs, we showed close alignment between training and generated sets (top 15 motifs: 0.1189 vs 0.1143 wrt average probability; bottom 15: 0.0189 vs 0.0160), confirming no significant bias. Additionally, we showed that **the marginal distributions of node/supernode categories remain consistent upon running DMol**.

We appreciate Reviewer FMwP's suggestion regarding FragFM but note our fundamentally different approach and a lack of codes for the method.

---

### Decision · Program_Chairs · 2025-09-17

**Decision:**

Accept (poster)

**Comment:**

This submission presents a new graph diffusion model, DMol, for small drug molecule generation, offering significant efficiency improvements over the DiGress method. While the reviewer pointed out some weaknesses, the overall assessment is positive. The model accelerates the diffusion process and improves the validity of generated graphs by sampling subgraphs at each step and modifying only the atoms and bonds within them, as well as by introducing an approximation that replaces ring structures with supernodes. Although there were concerns about the frequency of substructures related to the latter, the authors provided sufficient clarification in their rebuttal, which the reviewers acknowledged, leading to a meaningful discussion.

Overall, despite some remaining weaknesses, the strengths of the paper clearly outweigh the limitations. Therefore, I support accepting this paper.